JCB | Journal of Cell Biology

# REPORT

# Phosphoproteomic identification of Mos–MAPK targets in meiotic cell cycle and asymmetric oocyte divisions

Ivan Avilov[1], Yehor Horokhovskyi[2], Pooja Mehta[1], Luisa Welp[3], Jasmin Jakobi[1], Mingfang Cai[1], Aleksander Orzechowski[1], Henning Urlaub[3,4], Juliane Liepe[2], and Peter Lenart[1]

**The Mos kinase activates the ERK/MAPK pathway during oocyte meiosis, controlling essential meiotic functions in species across metazoa. However, despite its significance, the molecular targets of Mos–MAPK remain largely unidentified. Here, we addressed this question using starfish oocytes ideally suited to combine cellular assays with phosphoproteomics. This revealed CPE-mediated mRNA polyadenylation as a prominent target of Mos–MAPK, and we show that translation is required to drive the second meiotic division. Secondly, we identify a well-defined subset of cytoskeletal regulators as targets of Mos–MAPK. We show that this regulation is critical to ensure the asymmetry of meiotic divisions, primarily by reducing the growth of astral microtubules. This allows positioning of the spindle directly beneath the cortex and prevents the separation of spindle poles in anaphase, thereby minimizing polar body size. Thus, by phosphoproteomics, we reveal molecular modules controlled by Mos–MAPK, explaining how this single, conserved kinase can act as a switch between the mitotic and meiotic division programs.**

## Introduction

Oocyte meiosis is a specialized form of cell division adapted to produce the fertilizable egg. This is achieved through two highly asymmetric divisions, meiosis I (MI) and meiosis II (MII), without an intervening S phase, resulting in a single, large, haploid egg and small unviable polar bodies as byproducts (Mullen et al., 2019; Jessus et al., 2020). Strikingly, after fertilization, in the same egg cytoplasm, embryonic mitoses take place with alternating S and M phases, showing typical mitotic and lacking meiosis-specific features. What mediates this dramatic switch between meiotic and mitotic division programs?

The Mos kinase is a constitutive activator of the ERK/MAPK cascade that is present during oocyte meiosis and is degraded after fertilization (Dupré et al., 2010). Therefore, the ERK/MAPK pathway and the downstream effector p90[RSK] are active during oocyte meiosis, but not in mitosis. That this peculiar Mos–MAPK activity plays critical roles in driving the meiotic division program has been shown in species across metazoa: first in *Xenopus* oocytes (Sagata et al., 1988), followed by studies in mice (Colledge et al., 1994; Hashimoto et al., 1994), starfish (Tachibana et al., 2000), tunicates (Sensui et al., 2012), and cnidarian jellyfish (Amiel et al., 2009). In some of these species, Mos–MAPK

is required for triggering meiotic maturation (Sagata et al., 1988); however, a more widely conserved function of Mos–MAPK appears to be preventing S phase between MI and MII and driving oocytes directly into MII. Mos–MAPK inhibition causes oocytes to reform the nucleus and initiate S phase after MI in *Xenopus* and starfish oocytes (Sagata et al., 1988; Tachibana et al., 2000), and mouse oocytes also display signs of meiotic exit after MI (Araki et al., 1996; Phillips et al., 2002; Verlhac et al., 1996). An even more prominent function of Mos is to establish the "cytostatic factor (CSF) arrest," i.e., to halt meiosis until fertilization, and thereby synchronize these two vital processes (Jessus et al., 2020). Across species, Mos–MAPK inhibition leads to the loss of CSF arrest and parthenogenetic activation of oocytes (Sagata et al., 1988; Colledge et al., 1994; Hashimoto et al., 1994; Tachibana et al., 2000; Mori et al., 2006). Common to these seemingly unrelated functions is that they directly or indirectly involve increasing or sustaining Cdk1–cyclin B activity. While in most species the detailed mechanisms remain unclear, it is well understood in *Xenopus* oocytes that Mos–MAPK facilitates cyclin B translation to trigger meiotic maturation and to drive MII (Abrieu et al., 2001; Hochegger et al., 2001). Furthermore, for

[1]Research Group Cytoskeletal Dynamics in Oocytes, Max Planck Institute for Multidisciplinary Sciences, Göttingen, Germany;   [2]Research Group Quantitative and Systems Biology, Max Planck Institute for Multidisciplinary Sciences, Göttingen, Germany;   [3]Research Group Bioanalytical Mass Spectrometry, Max Planck Institute for Multidisciplinary Sciences, Göttingen, Germany;   [4]Cluster of Excellence "Multiscale Bioimaging: from Molecular Machines to Networks of Excitable Cells" (MBExC), University of Göttingen, Göttingen, Germany.

Correspondence to Peter Lenart: peter.lenart@mpinat.mpg.de.



maintenance of the CSF arrest, degradation of cyclin B is prevented by phosphorylation of Erp1/Emi2 by p90[RSK] downstream of Mos–MAPK (Wu and Kornbluth, 2008; Isoda et al., 2011).

Besides meiosis-specific cell cycle regulation, Mos–MAPK activity is also required for the asymmetric meiotic divisions. From jellyfish through starfish to mice, Mos–MAPK inhibition alters meiotic spindle morphology, converting them to somewhat "mitotic-like," displaying long, astral-like microtubules. This interferes with asymmetric division, resulting in enlarged polar bodies or failed polar body cytokinesis (Araki et al., 1996; Verlhac et al., 1996; Bodart et al., 2002; Amiel et al., 2009; Ucar et al., 2013). The reverse effect has also been observed in tunicates: mitotic embryos ectopically expressing Mos show short, meiotic-like spindles and extrude polar bodies (Dumollard et al., 2011). In mouse oocytes, two spindle regulators under the control of Mos–MAPK, MISS, and DOC1R have been identified and characterized in substantial detail (Lefebvre et al., 2002; Terret et al., 2003). However, a mechanistic model of how Mos–MAPK regulates spindle organization is lacking.

Here, we took advantage of starfish oocytes that are transparent, synchronous, and available in large quantities to identify downstream targets of Mos–MAPK by quantitative phosphoproteomics. We used imaging assays to characterize cellular phenotypes of Mos–MAPK inhibition in live oocytes and then analyzed samples by phosphoproteomics precisely timed to key transitions. This revealed conserved molecular modules under the control of Mos–MAPK, explaining how Mos–MAPK activity may drive the meiotic division program.

## Results and discussion

### Mos–MAPK drives the second meiotic division

First, we confirmed earlier observations by Tachibana and coworkers (Tachibana et al., 2000; Ucar et al., 2013) showing that the Mos kinase activates the ERK/MAPK pathway specifically during meiosis, is not active in immature oocytes, and is inactivated again at the end of MII (Fig. 1 A). This activity is effectively and specifically inhibited by 20 µM of the small-molecule MEK inhibitor, U0126, without interfering with hormone-induced maturation and the initial activation of Cdk1–cyclin B leading up to nuclear envelope breakdown (NEBD) (Fig. 1 A; and Fig. S1, A and B).

We then imaged live oocytes by high-resolution confocal microscopy to complement previous, detailed biochemical characterization of Mos–MAPK inhibition (Tachibana et al., 2000; Mori et al., 2006). After NEBD, chromosomes congressed to the animal pole both in U0126- and DMSO-treated oocytes, and the MI spindle formed with normal timing, indicating that these early steps of meiosis are independent of Mos–MAPK (Fig. 1, B and C). However, at metaphase the MI spindle showed a slightly enlarged morphology, and the polar bodies were consistently larger in U0126-treated oocytes (Fig. 1 B and Fig. S1, C–E and see below). Additionally, while DMSO-treated oocytes rapidly progressed to MII, forming the MII spindle within ~30 min, U0126-treated oocytes failed to transition to MII (Fig. 1 B). Unlike control oocytes, in which chromosomes remained condensed and the nuclear envelope did not reform

during the MI-MII transition, U0126-treated oocytes exited meiosis after MI, evidenced by decondensation of chromatin and reformation of the nucleus (Fig. 1, B and E). Even in the absence of fertilization, U0126-treated oocytes then initiated the first mitotic division and began parthenogenetic embryo development with similar timing to untreated and fertilized oocytes (Fig. 1, B and D). The second metaphase spindle, counting from the start of meiosis, in U0126-treated oocytes showed a morphology very similar to control mitotic spindles, about twice larger in length and centrosomal area size as meiotic spindles (Fig. S1, C–E).

In *Xenopus*, S phase is prevented between MI and MII by rapid reactivation of Cdk1–cyclin B, forcing direct entry into MII (Gerhart et al., 1984). This results from translation of cyclin B protein and activatory dephosphorylation of Cdk1 shortly after completion of MI (Hochegger et al., 2001; Iwabuchi et al., 2000). To test whether in starfish oocytes active Cdk1–cyclin B is sufficient to rescue the effect of Mos–MAPK inhibition, we injected oocytes with active Cdk1–cyclin B protein shortly after metaphase I (Fig. 1 F). This treatment reversed the effect of Mos–MAPK inhibition: while U0126-treated oocytes exited MI, formed a pronucleus, and entered parthenogenetic development, Cdk1–cyclin B-injected oocytes assembled an MII spindle without forming a pronucleus (Fig. 1 F). MII spindles in these Cdk1–cyclin B-injected oocytes showed a meiotic morphology, unlike the mitotic-like morphology in U0126-treated oocytes.

Taken together, consistent with previously published biochemical data, we show that in starfish oocytes Mos–MAPK is activated after NEBD, and its inhibition has no effect on maturation and spindle assembly before the metaphase of MI. However, soon after Mos–MAPK is activated, phenotypes of Mos–MAPK inhibition become evident: the metaphase I spindle is enlarged, resulting in the formation of an enlarged polar body. Subsequently, Mos–MAPK inhibition prevents reactivation of Cdk1–cyclin B after MI that would normally drive transition to MII. Therefore, Mos–MAPK-inhibited oocytes exit the meiotic program after MI and proceed to parthenogenetic development while skipping MII completely.

### Mapping downstream targets of Mos–MAPK by phosphoproteomics

Based on the above observations and taking advantage of the high synchronicity of starfish oocytes, we collected samples for phosphoproteomics at the metaphase of MI; soon after Mos–MAPK is activated, and as soon as the first cellular phenotypes of Mos–MAPK inhibition are evident. Thereby, we expect to identify mostly direct effects. Oocytes were collected from three starfish animals, each treated with U0126 or an equal amount of DMSO and then matured (Fig. 2 A). Synchronicity was confirmed by staining a small portion of the sample with Hoechst 33342, which showed 92–96% metaphase figures. Samples were then enriched for phosphopeptides and analyzed by multiplexed quantitative tandem mass tag (TMT) mass spectrometry, identifying 9,035 protein groups, of which 3,197 contained phosphopeptides (see Data S1 for complete dataset).

We then used ratios of phosphorylated peptides to their total protein intensities in U0126- vs. DMSO-treated samples (of

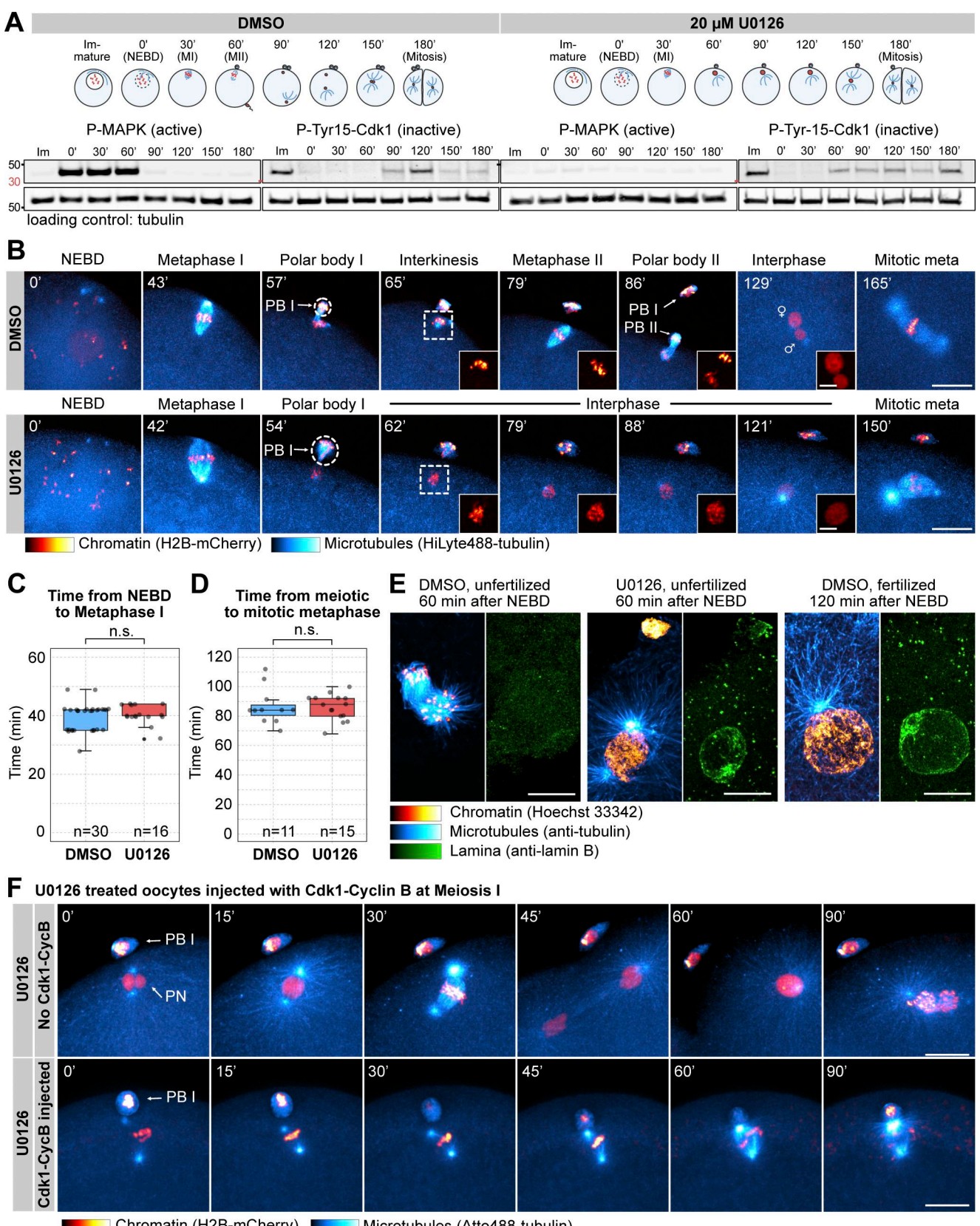

Figure 1. **Mos–MAPK inhibition prevents MII leading to parthenogenetic development. (A)** Western blot of a time course using phospho-MAPK (marking activated MAPK) and phospho-Cdk1 (marking the inactive Tyr15 phosphorylated form) antibodies to monitor Mos–MAPK and Cdk1 activities during meiosis of starfish oocytes. Oocytes were treated with 20 μM U0126 or an equal amount of DMSO added simultaneously with the maturation hormone. Time points are indicated as minutes after NEBD. Fertilization was performed 60 min after NEBD. Numbers to the left are in kDa and mark the position of the molecular weight

markers. For the complete blots see Source Data F1. **(B)** Selected frames from time-lapse recording of oocytes expressing H2B-mCherry (red) and injected with HiLyte488-tubulin (cyan) to label chromosomes and microtubules, respectively. Oocytes were imaged through meiosis and then fertilized after MI while imaging on a spinning disk confocal microscope. Before inducing maturation, oocytes were treated either with 20 μM U0126 or an equal amount of DMSO. PBI and outlines indicate the polar bodies. Venus and Mars signs indicate female and male pronuclei, respectively. Insets show magnified views of chromosomes. Maximum intensity projections are shown, scale bars are 20 μm (insets 5 μm), and time is given in minutes. **(C)** Individual data points and a boxplot showing the time from NEBD to metaphase I quantified on recordings similar to those shown in B. Statistical comparison was done using the Mann–Whitney test; n marks the number of oocytes imaged. **(D)** Individual data points and a boxplot showing the time from meiotic metaphase I to the first mitotic metaphase quantified on recordings similar to those shown in B. Statistical comparison was done using the Mann–Whitney test, n marks the number of oocytes imaged. **(E)** Immunofluorescence images of oocytes stained for microtubules (anti-tubulin, cyan), chromatin (Hoechst 33342, red), and the lamina (anti-lamin B, green). Oocytes were treated with 20 μM U0126 or an equal amount of DMSO and then fixed 60 and 120 min after NEBD, respectively. Samples were imaged on a spinning disk microscope. Maximum intensity projections are shown; scale bars are 10 μm. **(F)** Selected frames from a time-lapse recording of oocytes expressing H2B-mCherry (red) and injected with HiLyte488-tubulin (cyan) to label chromosomes and microtubules, respectively. Oocytes were imaged through meiosis on a spinning disk confocal microscope. Before inducing maturation, oocytes were treated with 20 μM U0126. Then oocytes were either left unmanipulated, or they were injected with active Cdk1–cyclin B protein immediately following extrusion of the first polar body. Maximum intensity projections are shown, scale bars are 20 μm, and time is given in minutes. PB I denotes the first polar body, and PN is the pronucleus. Source data are available for this figure: SourceData F1.

14,360 phosphopeptides identified, 10,880 unique phosphopeptides and 16,621 sequence-PTM combinations were quantified). This resulted in 211 unique phosphopeptides (205 down-, 6 up-regulated, and 360 sequence-PTM combinations) that passed significance threshold (Data S3). Unsupervised clustering clearly separated U0126-treated samples from DMSO controls, evidencing an effect much above biological variability despite sampling genetically diverse individuals (Fig. 2 B).

Next, we performed an overrepresentation analysis revealing the enrichment of gene ontology (GO) terms related to cell cycle and cytoskeletal regulation among the dephosphorylated proteins (Fig. 2 C). We also predicted putative human protein kinases corresponding to the observed dephosphorylation activity (Johnson et al., 2023). This showed an enrichment of kinases of the MAPK pathway (STE kinase family) and its downstream effector p90[RSK] (AGC kinase family) (Fig. 2 D). Analysis of the predicted upstream kinases revealed that the Mos–MAPK–p90[RSK] pathway had one of the largest sets of peptide targets and highest predicted ranks, followed by putative targets of Cdk1 (Fig. S2 A).

We complemented the phosphoproteomics experiment at metaphase of MI with a data-independent acquisition (DIA) proteomics experiment to monitor changes in protein levels at subsequent stages (Fig. 2 E, see Data S2 for the complete dataset). In this experiment we were able to identify 7,480 protein groups, of which 5,103 protein groups could be quantified. This dataset did not reveal major overall changes in protein abundance consistent with previous work by Swartz et al. (2021), showing that the proteome is largely stable during meiosis in starfish oocytes (Fig. 2 F) (Swartz et al., 2021). However, they showed that a very limited subset of proteins, including key cell cycle regulators, do change in abundance, which we were able to recapitulate in our time course experiment (Fig. 2 G and Fig. S2 B and see discussion below).

### Phosphoproteomics identifies functional modules downstream of Mos–MAPK

We then displayed the differentially phosphorylated peptides on a volcano plot and manually annotated these hits to identify functional modules (Fig. 3, A and B; Table S1; and Data S3).

In the U0126-treated samples, we detected dephosphorylation of MEK and the downstream effector p90[RSK], consistent with effective inhibition of the Mos–MAPK pathway. Additionally, we found several components of the core cell cycle machinery to be differentially dephosphorylated, including the Cdc25 phosphatase, the Myt1 kinase, and the APC/C subunit Cdc27 (APC3), as well as Arpp19 and PP2A. These changes indicate that Mos–MAPK inhibition acts toward Cdk1–cyclin B inactivation, suggesting that normally Mos–MAPK activity contributes to keeping Cdk1 activity high to allow rapid transition to MII—fully consistent with our findings above.

Manual annotation revealed translational regulation as another prominent functional module downstream of Mos–MAPK. This was not immediately apparent in the GO-term enrichment analysis, as we only identified a specific subset of 11 out of the 1,059 proteins associated with the broad term "translation." These proteins are components of the CPE-mediated mRNA polyadenylation pathway that is known to have critical importance in oocytes documented extensively in *Xenopus* as well as in other model systems (Conti and Kunitomi, 2024; Richter and Lasko, 2011; Mendez and Richter, 2001; Piqué et al., 2008). CPEB binds to the CPE element in the 3′UTR of transcripts and forms both repressive complexes that prevent translation and the active complex that facilitates translation by polyadenylation of the mRNA. During oocyte maturation, CPEB is thought to be first phosphorylated by Aurora A and then hyperphosphorylated by Cdk1 and Mos–MAPK. These phosphorylations are required for the formation of the active complex and lead to subsequent degradation of CPEB (Mendez et al., 2000a; Thom et al., 2003; Keady et al., 2007; Lapasset et al., 2008; Ochi et al., 2016). The potential Aurora site is conserved in starfish, but we could not detect phosphorylation of this site. Instead, we detected the largest change on Ser32, a site not well conserved but located near the Thr22 site, a target of Mos–MAPK in *Xenopus* oocytes (Keady et al., 2007). We detected additional CPEB phosphopeptides consistent with hyperphosphorylation by Cdk1 and Mos–MAPK. The proteomics time course experiment also confirmed that while CPEB is still detectable at MII, levels gradually decrease both in control and U0126-treated samples (Fig. 2 G). Interestingly, CPEB levels at MI appear somewhat lower in U0126-treated samples as compared with controls; that could explain, at least in part, the lower level of cyclin B in this condition.

We detected dephosphorylation of the eIF4E transporter, eIF4ENIF1, a.k.a. 4E-T, which, along with CPEB, is a core

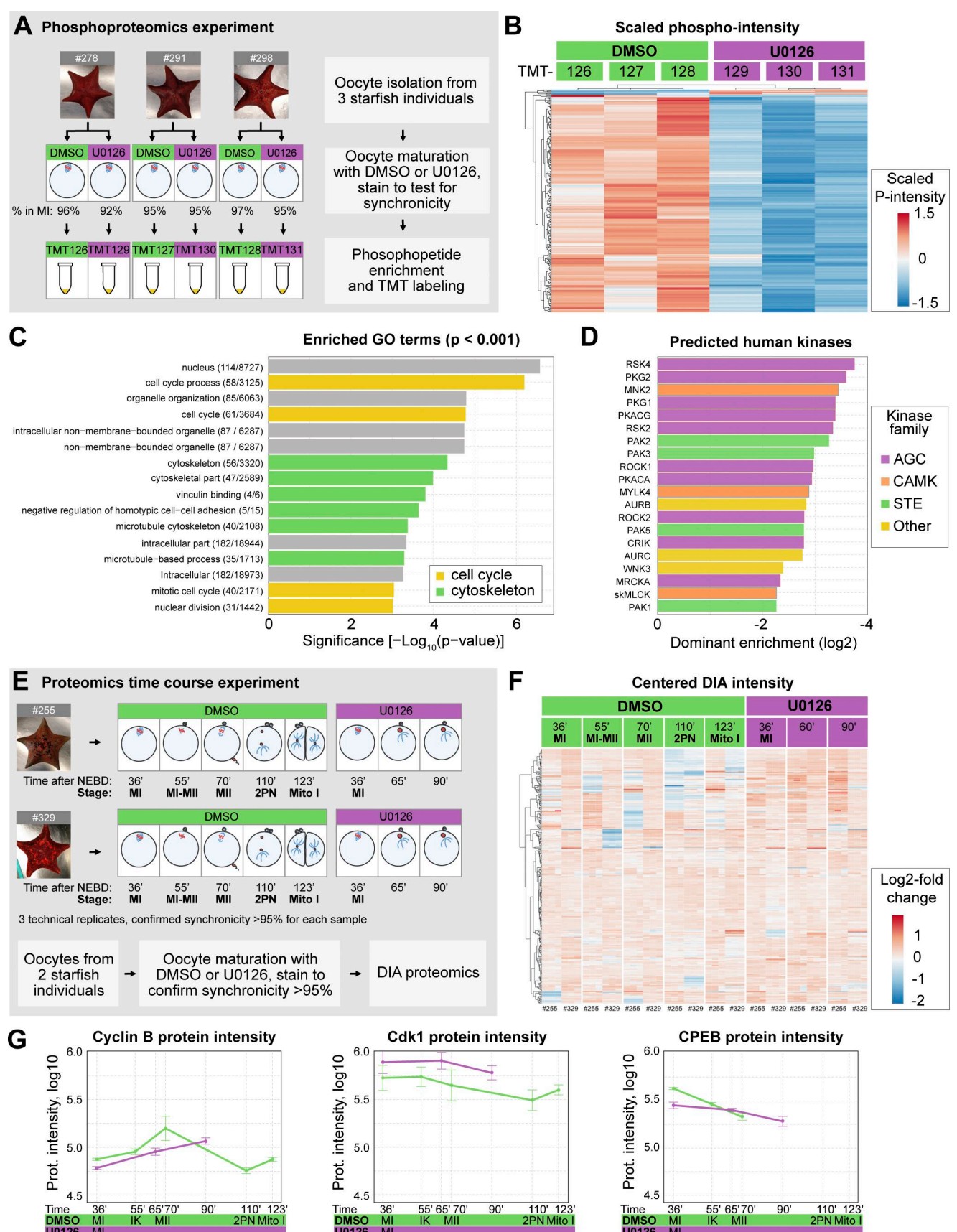

Figure 2. **Overview of the proteomics and phosphoproteomics experiments. (A)** Outline of the phosphoproteomics experiment. Mos–MAPK-inhibited (20 µM U0126, purple) and control (DMSO, green) oocytes were collected at metaphase I in three biological replicates and processed for phosphoproteomics. For

Phosphoproteomics identifies targets of Mos–MAPK

the complete dataset, see Data S1. **(B)** Unsupervised clustering of differentially phosphorylated peptides (rows) vs TMT channels (columns). Green: control, purple: Mos–MAPK inhibition. TMT reporter intensity values are scaled to a mean of 0 and SD of 1. **(C)** GO overrepresentation analysis of differentially phosphorylated proteins. The numbers indicate the gene set sizes and a number of differentially phosphorylated proteins. **(D)** Prediction of human effector kinase families from the differentially dephosphorylated sites. **(E)** Outline of the proteomics time course experiment. Mos–MAPK-inhibited (20 μM U0126, purple) and control (DMSO, green) oocytes were collected at the specified stages at the given times in two biological replicates. Samples were processed in three technical replicates for proteomics and analyzed by DIA. For the complete dataset, see Data S2. **(F)** Unsupervised clustering of centered DIA intensities of two biological and three technical replicates (columns). Green: control, purple: Mos–MAPK inhibition. **(G)** $Log_{10}$ normalized protein intensities for the indicated proteins. The mean and standard error of the two biological replicates and three technical repetitions of each are shown.

---

component of the repressive ribonucleoparticle (RNP) keeping mRNAs translationally inactive. We detected peptides of other components of RNPs, including eIF4E, DDX6, and LSM14 (but not Zar1) (Minshall et al., 2007; Cheng and Schuh, 2024). These proteins are present at constant levels through meiosis, and we saw no significant change in their phosphorylation (Fig. 3 B and Fig. S2 B). We also detected many phosphopeptides of TACC3/maskin, and while the changes remain below significance, they show a trend toward dephosphorylation (Data S1). This suggests that TACC3/maskin may play a similar role as in *Xenopus* oocytes and interact with eIF4E to regulate translation (Stebbins-Boaz et al., 1999).

Upon CPEB phosphorylation, the repressive complex dissociates, and so CPEB can interact with the multi-subunit cleavage and polyadenylation specificity factor to recruit the polyA polymerase PAP/Gld2 (Mendez et al., 2000b; Barnard et al., 2004). Besides Gld2, we observed a change in phosphorylation of several components of the cap-dependent translation initiation complex, including the scaffold protein eIF4G that binds to the polyadenylate-binding protein PABPC on the semicircularized mRNA (Wells et al., 1998) and eIF3A involved in the eventual recruitment of the small ribosomal subunit (Aitken and Lorsch, 2012) (Fig. 3 B and Table S1). Interestingly, the nonessential initiation factor eIF4B showed increased phosphorylation upon U0126 treatment. While we did not detect a change in phosphorylation, we did detect peptides of most other subunits of the complex, such as the cap-binding protein eIF4E, the helicase eIF4A, the polyadenylate-binding protein PABPC (Fig. S2 B).

Interestingly, our top hits also included LARP1, a regulator specific to mRNAs with a 5′ terminal oligopyrimidine (5′TOP), a motif known to associate with mTOR targets (Berman et al., 2021). LARP1 is thought to bind this 5′TOP motif and stabilize the semicircularized mRNA through interaction with PABPC (Hochstoeger et al., 2024).

Another large group of phosphopeptides identified were regulators of the cytoskeleton, a substantial subset of which are involved in centrosomal microtubule nucleation (Fig. 3, A and B; and Table S1). We found components with roles in centrosome maturation, including proteins involved in the recruitment of the pericentriolar material (PCM) to centrioles (CEP192, CEP44, and POC1), the main PCM constituent PCM1, as well as the protein furry-like, thought to be important for centrosome integrity early in cell division (Ikeda et al., 2012; Conduit et al., 2015). Secondly, we found several proteins that play a role in nucleating and stabilizing astral microtubules, including the general microtubule plus tip-binding proteins XMAP215/chTOG and CLIP1 (Slep and Vale, 2007), and GTSE1, a plus tip-binding

protein that specifically regulates the stability of astral microtubules (Bendre et al., 2016). Furthermore, we identified several members of the kinesin superfamily. A kinesin-4 family member (identified as KIF4/21/27) may be involved in separation of spindle poles in anaphase (Bieling et al., 2010), and the kinesin-3 KIF14 may be required for anaphase in interaction with citron kinase (Gruneberg et al., 2006). Interestingly, a meiosis-specific paralog of KIF14, NabKin, was reported to regulate polar body cytokinesis in *Xenopus* oocytes (Samwer et al., 2013).

We also identified several proteins that regulate the actin cytoskeleton (Fig. 3, A and B; and Table S1). In particular, regulators of the small GTPases Cdc42, Rac1, and RhoA, as well as their downstream effectors, have been detected. This includes BCR, which is a GTPase-activating protein for Cdc42, as well as the activator of RhoA, MYO9. We detected a significant dephosphorylation of the formin mDia2, which confirms earlier results demonstrating that mDia2 is a downstream target of Mos–MAPK in starfish (Ucar et al., 2013). Additionally, we found a significant effect on the Cdc42-binding protein MRCK, which recently has been shown to drive the initial, meiosis-specific step of polar body cytokinesis (Bourdais et al., 2023). Finally, we found Rac1, which may be involved in Arp2/3-dependent actin nucleation and act as an antagonist of RhoA during polar body cytokinesis, to be dephosphorylated (Halet and Carroll, 2007; Pal et al., 2020).

Taken together, our phosphoproteomic analysis identified well-defined and conserved molecular modules downstream of Mos–MAPK that are involved in the regulation of the meiotic cell cycle, CPE-dependent translation, and the microtubule and actin cytoskeleton. Firstly, these data are consistent with the model proposed in *Xenopus*, whereby Mos–MAPK drives MI-MII transition both by facilitating rapid reactivation of Cdk1 by its dephosphorylation, as well as by triggering translation of cyclin B and other regulators (Hochegger et al., 2001; Iwabuchi et al., 2000). Our data support a possible model for translational regulation, by which Mos–MAPK and Cdk1 phosphorylate CPEB, leading to the disassembly of repressive RNPs and promoting the assembly of the active polyadenylation complex.

Secondly, Mos–MAPK regulates the cytoskeleton to mediate the highly asymmetric oocyte divisions. We identified microtubule regulators, including constituents of the centrosome, the PCM, regulators of microtubule growth at the plus tips of microtubules, and kinesin motor proteins. These proteins are not part of a single complex, but they do have a common function in organizing centrosomal microtubule arrays. These hits are distinct from regulators of other major pathways of spindle assembly, such as the chromatin-mediated Ran pathway (Valdez et al., 2023). Interestingly, a very recent study found that casein

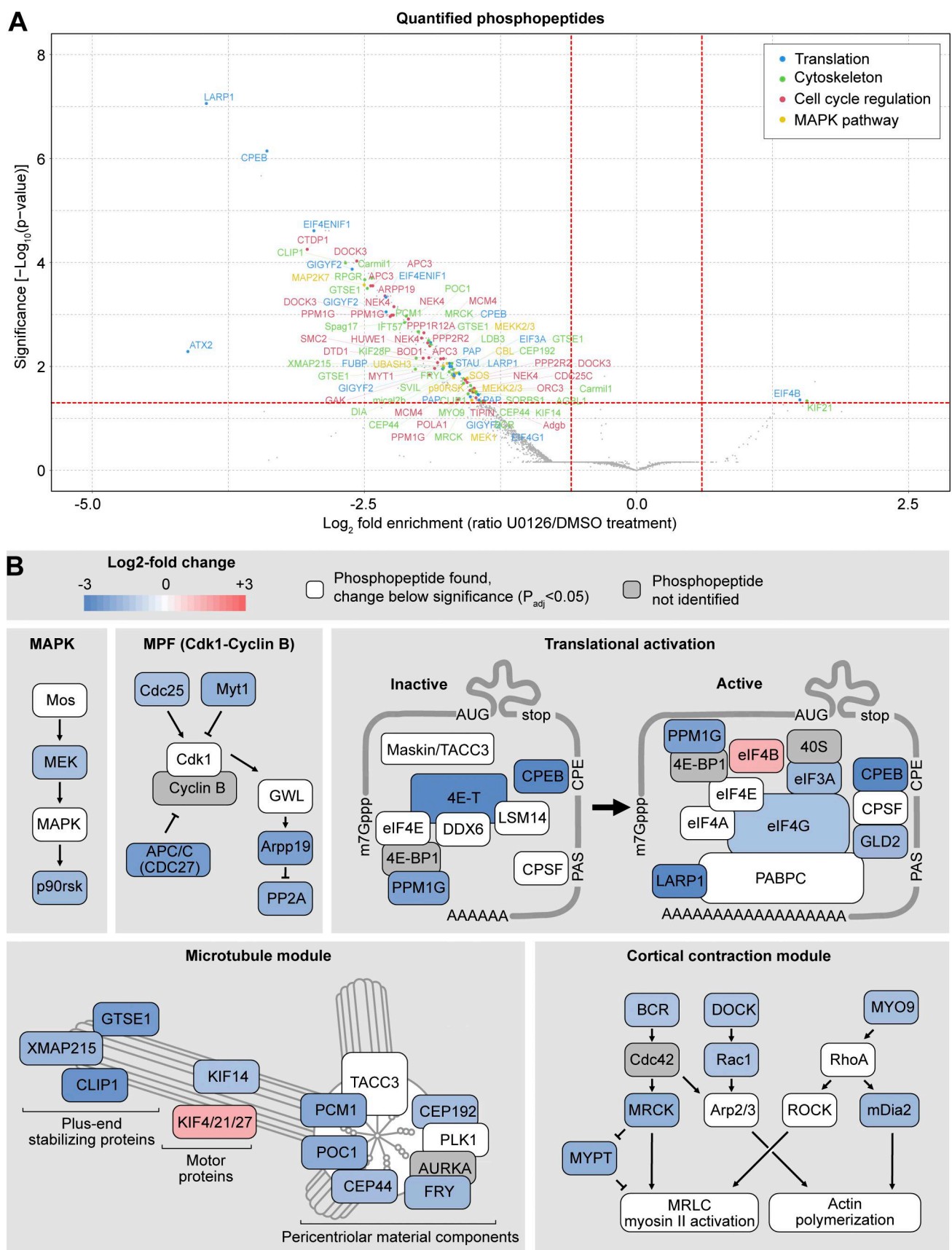

Figure 3. **Phosphoproteomics reveals cell cycle, cytoskeleton, and translation as the main processes regulated by Mos–MAPK. (A)** Volcano plot of the quantified phosphopeptides. The majority of phosphopeptides were manually classified into four categories: translation, cytoskeleton, cell cycle regulation, and

MAPK pathway. The rest of the phosphopeptides, including those from unidentified proteins, are marked as gray dots. For the complete dataset, see Datas S1 and S3. **(B)** Functional modules were manually assembled from the list of significantly affected phosphopeptides. Color code indicates the average $\log_2$ fold change in phosphopeptide abundance upon Mos–MAPK inhibition. The numerical data are shown in a tabulated form on Table S1.

kinase 2 specifically regulates the Ran pathway in meiosis (Cantwell et al., 2025). This suggests that there may be multiple kinases regulating different aspects of meiotic spindle assembly, with Mos–MAPK specifically acting on astral microtubule arrays.

**Mos–MAPK prevents assembly of astral microtubule arrays**
To explore the effects of Mos–MAPK on the cytoskeleton in more detail, we recorded the dynamics of MI spindles in 3D at high resolution in live oocytes (Fig. 4 A). Quantification of the maximal length of the astral microtubules and the distance between the spindle poles revealed that U0126-treated oocytes display longer astral microtubules and increased pole-to-pole distance. The difference is initially subtle, but it then increases continuously as oocytes progress from prometaphase to telophase (Fig. 4, A–C). Indeed, Mos–MAPK qualitatively changes the behavior of spindle poles: while in control oocytes with Mos–MAPK active astral microtubules and pole-to-pole distance shorten, in U0126-treated oocytes, asters and pole-to-pole distance increase as spindles progress toward telophase (Fig. 4, A–C). As a consequence, the first polar body nearly doubles in size, and polar body cytokinesis more frequently fails in Mos–MAPK-inhibited oocytes (Fig. 4, D and E).

Next, we performed immunofluorescence stainings to visualize spindle organization at even higher resolution (Fig. 4 F). These data corroborated observations made in live oocytes: in normal meiosis centrosomes are almost completely inactive, and chromosomes separate merely by shortening of kinetochore fibers (anaphase A) (Fig. 4, F–H). By stark contrast, Mos–MAPK-inhibited oocytes display prominent microtubule asters nucleated by centrosomes, which contain numerous and long microtubules (Fig. 4, F–H). U0126-treated spindles also display a prominent mid-spindle formed by overlapping microtubules, which drive separation of spindle poles concomitant with shortening of kinetochore fibers in anaphase (anaphase A+B) (Fig. 4 F).

Taken together, in excellent agreement with the identified phosphoproteins, we show that in unperturbed MI Mos–MAPK activity impedes centrosomal microtubule nucleation and prevents formation of the spindle midzone and anaphase B. If Mos–MAPK is inhibited, the MI spindle adopts a mitotic-like organization with large centrosomal asters and a prominent mid-spindle promoting separation of poles in anaphase B. As a consequence, enlarged polar bodies form, and polar body cytokinesis frequently fails. Intriguingly, our data reveal a tight correlation between the maximal length of astral microtubules and pole-to-pole distance, suggesting that the two processes may be functionally linked (Fig. 4, B and C, see schematics on Fig. 5 E). The explanation for this can be that centrosomal microtubules grow in all directions: away from the spindle as astral microtubules and into the spindle to form interpolar microtubules. Interpolar microtubules are critical to form the overlapping

microtubule bundles at the spindle midzone, pushed apart in anaphase B by PRC1 and kinesin-4 (Bieling et al., 2010). In meiosis, with Mos–MAPK active, both astral and interpolar microtubules are absent, and thus anaphase B does not occur. Through these mechanisms Mos–MAPK functions to minimize polar body size in two ways: by reducing aster size it allows the spindle to localize tightly to the cortex, and by preventing anaphase B it limits the distance between the cortex and the spindle midzone (Fig. 5 E). Therefore, Mos–MAPK essentially switches off two mitotic spindle functions: positioning by astral microtubules and anaphase B that in mitosis is required to move chromosome masses away from the cleavage furrow.

In starfish oocytes, unlike in most other species, centrioles are still present at MI (Sluder et al., 1993; Borrego-Pinto et al., 2016a). This might have made the phenotypes easier to interpret than in other species with acentrosomal meiotic spindles. However, acentrosomal microtubule-organizing centers (aMTOCs) contain most if not all of the proteins we identified and functionally replace centriolar centrosomes in oocytes (Mogessie et al., 2018; Schuh and Ellenberg, 2007). Indeed, Mos–MAPK inhibition results in enlarged microtubule asters nucleated from aMTOCs in *Xenopus*, mouse, and *Clytia* oocytes (Verlhac et al., 1996; Bodart et al., 2002; Amiel et al., 2009), and in mouse spindle elongation, i.e., anaphase B, was observed in $Mos^{-/-}$ oocytes, very similar to our observations above (Verlhac et al., 2000). This suggests that Mos–MAPK likely has a conserved role in species both with centrosomal and acentrosomal MTOCs. In turn, this meiosis-specific spindle architecture may explain why oocytes employ specialized, actin-driven mechanisms for spindle positioning (Field and Lénárt, 2011).

**Translation is required to drive MII, but it does not affect spindle architecture**
While the observed effects on spindle organization could be fully explained by direct phosphorylation of proteins of the "microtubule module" we identified, alternatively, Mos–MAPK-dependent translation of specific microtubule regulators may also affect spindle architecture. To separate out these effects, we inhibited translation by emetine (Swartz et al., 2021). Emetine-treated oocytes progressed to MI without delay, assembled the MI spindle with apparently normal morphology, and extruded a polar body. Thereafter, a pronucleus formed, and oocytes remained in a sustained interphase arrest (Fig. 5 A). This confirms that in starfish oocytes new protein synthesis is not needed until the end of MI; however, new protein synthesis is required for progression to MII (Swartz et al., 2021).

Next, to investigate the effects of translation inhibition on spindle organization, we fixed and stained oocytes at metaphase of MI that were treated with emetine and with emetine and U0126 (Fig. 5 B). Emetine alone had no visible effect on spindle organization, whereas a combined treatment with emetine and U0126 resulted in the same phenotype with enlarged asters and

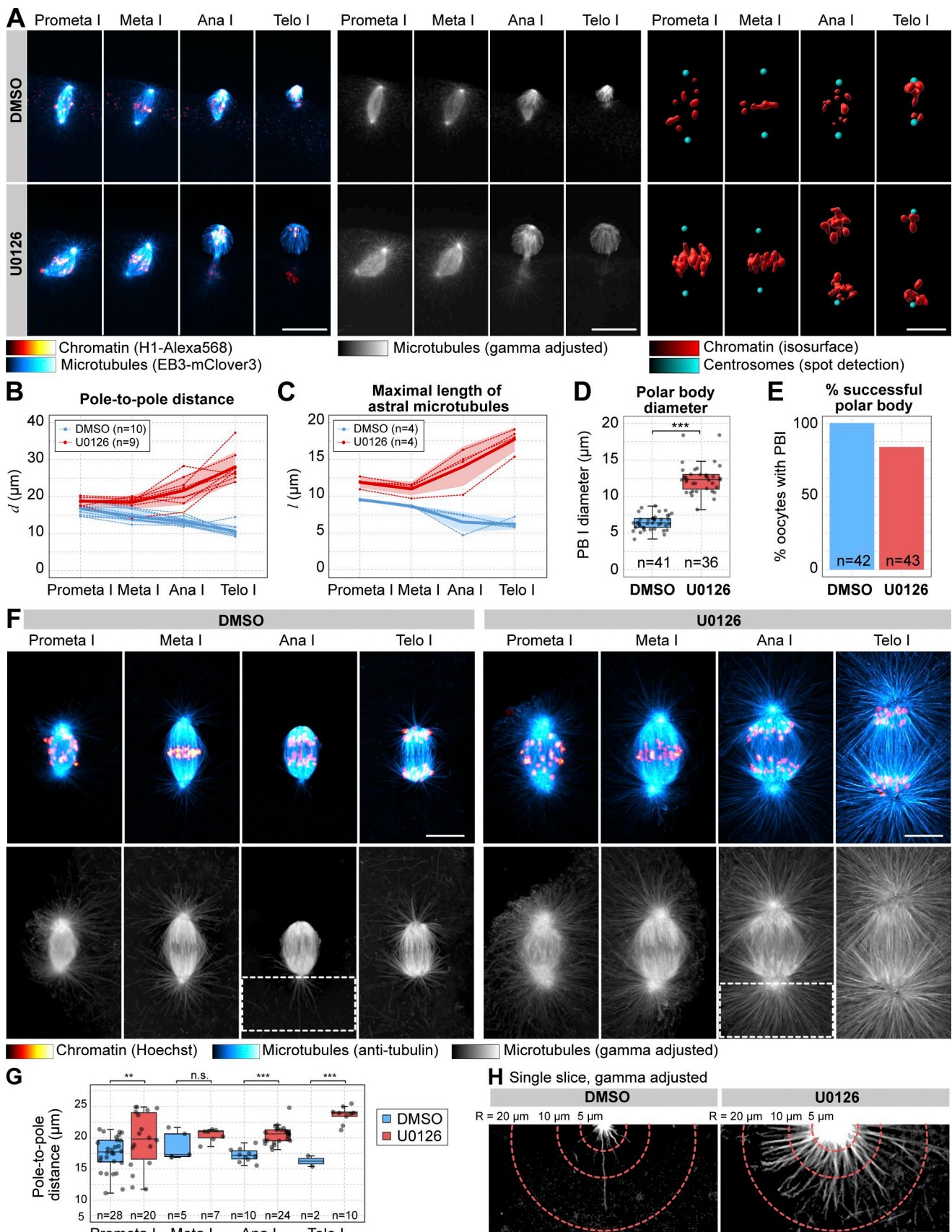

Figure 4. **Mos–MAPK reduces aster growth to minimize polar body size. (A)** Selected frames from a time-lapse recording of the microtubule spindle in an oocyte undergoing MI on a spinning disk confocal microscope. Chromosomes are labeled with H1-Alexa568 (red) and microtubule plus tips by EB3-mClover3

protein (cyan). Before inducing maturation, the oocyte was treated with either 20 µM U0126 or an equal amount of DMSO. (Left) overlay of the two fluorescent channels; (middle) the EB3-mClover3 channel is shown with adjusted gamma; (right) 3D rendering of the centrosome (cyan) and chromosomes (red). Maximum intensity projections are shown; scale bars are 20 µm. **(B and C)** 3D distances of the spindle poles and the maximal length of astral microtubules at different stages of MI are quantified on recordings similar to A. *n* refers to the number of individual oocytes analyzed. **(D)** Boxplot showing quantification of polar body sizes on transmitted light microscopy images. *n* refers to the number of individual oocytes analyzed. ***P < 0.0001, statistical comparison was done using a two-tailed *t* test. **(E)** Bar chart showing the success rate of polar body formation. Polar bodies were counted by live imaging in H1-Alexa568–injected oocytes after completion of meiosis. *n* refers to the number of individual oocytes analyzed. **(F)** Immunofluorescence images of oocytes stained for microtubules (anti-tubulin, cyan) and chromatin (Hoechst 33342, red). Oocytes were treated with 20 µM U0126 or an equal amount of DMSO and then fixed at the stages indicated. Imaging was done using spinning disk confocal microscopy. Maximum intensity projections are shown; scale bar is 20 µm. Lower panels show the tubulin channel separately and with adjusted gamma (0.5). **(G)** Quantification of the 3D distance between spindle poles on data similar to that shown in F. **P < 0.05, ***P < 0.0001 in two-tailed *t* tests. **(H)** Single optical sections crossing the centrosomes selected from the projected images shown in F. Images are shown after gamma adjustment (0.5); scale bar is 10 µm.

pole-to-pole distance as U0126 alone (compare Fig. 4, F and G; and Fig. 5, B–D). To specifically test whether translation of cyclin B—the protein the levels of which change to the largest extent during meiosis—has an effect on MI spindle organization, we injected oocytes with a morpholino targeting cyclin B. While the cyclin B morpholino caused an effect very similar to translation inhibition by emetine in preventing MII, it had no apparent effect on the morphology of the MI spindle (Fig. S3).

Thus, these data together show that in starfish oocytes protein synthesis is not required for meiotic maturation and meiotic progression until the end of MI; however, translation of cyclin B and likely other proteins is required to drive MII. Furthermore, our data show that Mos–MAPK acts on spindle organization independently of its effect on translational regulation.

## Conclusions

Mos–MAPK and its functions in oocyte meiosis have been studied extensively in the past. These studies evidence that Mos is an "early invention" already present in the non-bilaterian cnidarian jellyfish, and is thus an ancestral and conserved regulator of oocyte meiosis in metazoa (Amiel et al., 2009). However, likely due to the inherent diversity in cellular morphology and timing of oocyte meiosis among animal species, it remained difficult to identify conserved molecular mechanisms under the control of Mos–MAPK (Dupré et al., 2010). Here, by combining cellular assays with phosphoproteomics, we identified the molecular modules controlled by Mos–MAPK: translational regulation by CPE-mediated mRNA polyadenylation, and regulation of astral microtubule arrays. We show how these processes are critical for switching between mitotic and meiotic division programs for adapting the cell division machinery to produce the haploid egg ready to be fertilized.

## Materials and methods
### Oocyte collection, drug treatments, and microinjection
Bat stars (*Patiria miniata*) were obtained in the springtime from Southern California (South Coast Bio-Marine LLC, Monterey Abalone Company, or Marinus Scientific, Inc). They were kept at 16°C for the rest of the year in seawater aquariums at MPI-NAT's Animal Facility. Ovaries and testes were isolated as described previously (Borrego-Pinto et al., 2016b). Oocytes were isolated and used immediately or stored for 1–2 days in seawater at 14°C; for some experiments ovaries were stored for 2–3 days in

seawater at 14°C, and oocytes were isolated thereafter (Wessel et al., 2010). For fertilization, freshly isolated sperm was added just prior to the emission of the first polar body.

mRNAs and other fluorescent markers were injected using Kiehart chambers and horizontally mounted needles as described previously (Borrego-Pinto et al., 2016b; Jaffe and Terasaki, 2004), with the modification that instead of mercury-filled needles, simple back-loaded capillaries were used on a FemtoJet (Eppendorf) injector. mRNA was injected the day before to allow protein expression, whereas fluorescently labeled proteins were injected a few hours prior to imaging.

Meiosis was induced by the addition of 1-methyladenine (1-MA; Acros Organics) at 10 µM final concentration. NEBD started typically 20–25 min after 1-MA addition. U0126 (Sigma-Aldrich) was added simultaneously with 1-MA at 20 µM final concentration. Emetine (Sigma-Aldrich) was added at 10 µM final concentration 30 min before 1-MA.

Every experiment was repeated at least three times, with oocytes taken from at least two different animals.

### Fluorescent markers
Chromosomes were labeled by either injecting mRNA encoding human H2B-mCherry or H1 protein purified from calf thymus (Sigma-Aldrich) labeled with AlexaFluor 568 or 647 as described previously (Bun et al., 2018). Microtubules were labeled either by injection of HiLyte488- or HiLyte647-tubulin purified from porcine brain (Cytoskeleton) or recombinant human EB3-mClover protein (gift from So Chun, MPI-NAT). Anti–cyclin B morpholino (5′-TAACCAATGCGAGTTCCGAGGAG-3′, Gene Tools) was injected at 500 µM final concentration immediately before 1-MA stimulation (Swartz et al., 2021).

Capped mRNAs with poly-A tails were synthesized from linearized pGEMHE plasmids using the HiScribe T7 ARCA mRNA Kit (NEB). The mRNA was purified by the Monarch RNA Cleanup Kit (NEB).

The active Cdk1–cyclin B complex was a gift from Pim J. Huis In't Veld and Andrea Musacchio (MPI for Molecular Physiology, Dortmund, Germany) (Huis In 't Veld et al., 2022). To remove glycerol, the complex was purified through a 10-K MWCO protein concentrator (Pierce) and dissolved in PBS with 200 mM sucrose. The activity of the complex was confirmed by its ability to cause maturation in starfish oocytes. The Cdk1–cyclin B complex was injected into the U0126-treated oocytes directly after the first polar body extrusion.

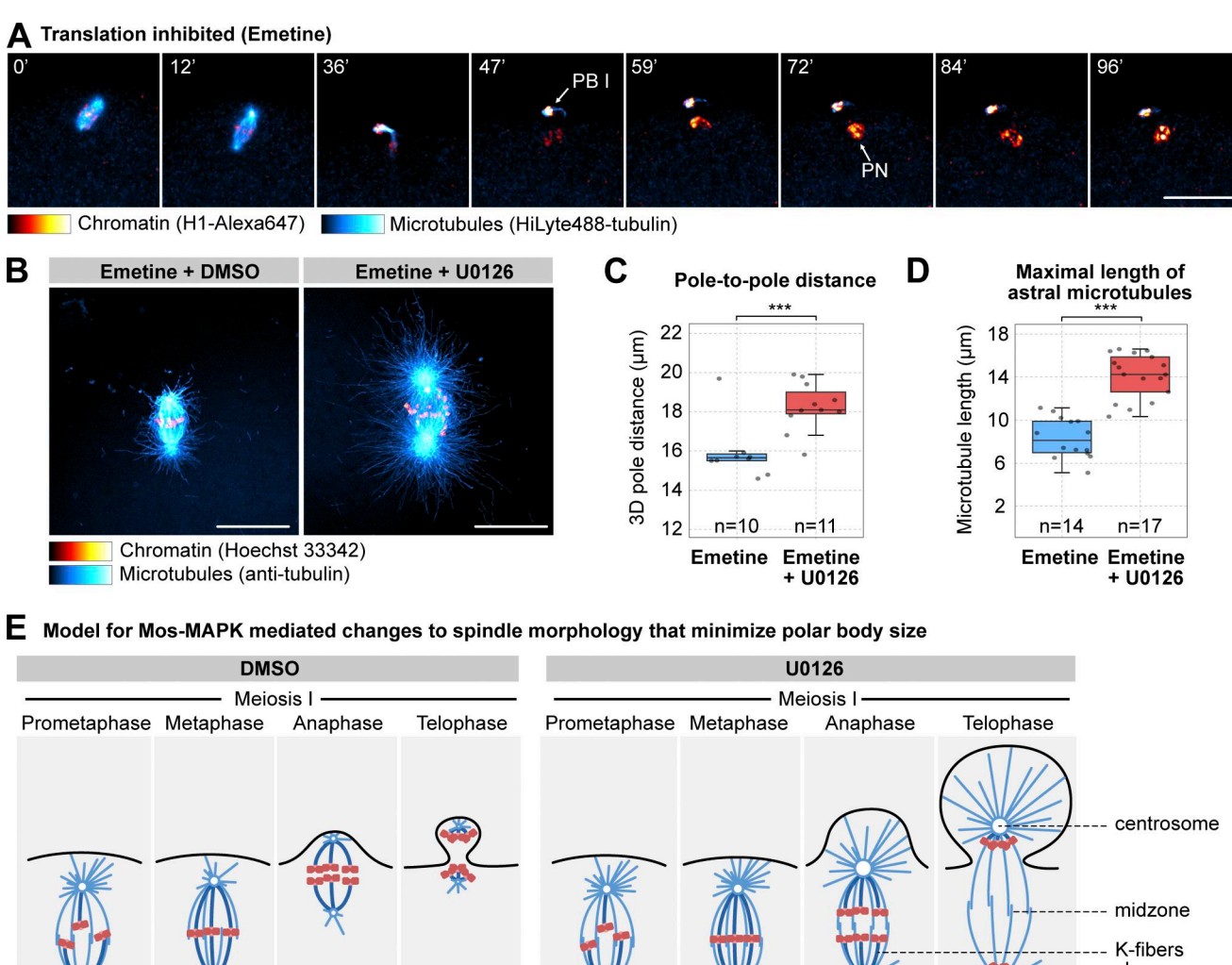

**A** Translation inhibited (Emetine)

Chromatin (H1-Alexa647)  Microtubules (HiLyte488-tubulin)

**B** Emetine + DMSO / Emetine + U0126

Chromatin (Hoechst 33342)
Microtubules (anti-tubulin)

**C** Pole-to-pole distance

3D pole distance (μm)

n=10 / n=11
Emetine / Emetine + U0126

**D** Maximal length of astral microtubules

Microtubule length (μm)

n=14 / n=17
Emetine / Emetine + U0126

**E** Model for Mos-MAPK mediated changes to spindle morphology that minimize polar body size

DMSO — Meiosis I — Prometaphase Metaphase Anaphase Telophase

U0126 — Meiosis I — Prometaphase Metaphase Anaphase Telophase

centrosome
midzone
K-fibers
chromosomes
astral microtubules

Average values taken from Fig. 3B and C: $l$ = length of astral microtubules  $d$ = pole-to-pole distance

Figure 5.  **Translation is required for MII transition, but it does not affect spindle morphology. (A)** Selected frames from a time-lapse recording of an oocyte injected with H1-Alexa647 (red) and HiLyte488-tubulin (cyan) to label chromosomes and microtubules, respectively. The oocyte was imaged through meiosis on a spinning disk confocal microscope. Before inducing maturation, the oocyte was treated with 10 μM emetine to block translation. Maximum intensity projections are shown, the scale bar is 20 μm, and time is given in minutes. PB I denotes the first polar body, and PN is the pronucleus. **(B)** Oocytes fixed at metaphase I and stained for microtubules (anti-tubulin, cyan), chromatin (Hoechst 33342, red). Oocytes were treated with emetine and DMSO or 20 μM U0126. 3D stacks were acquired on a spinning disk microscope. Maximum intensity projections are shown; scale bars are 20 μm. **(C and D)** Boxplots showing a comparison of the pole-to-pole distance and maximal length of microtubules at metaphase I between emetine and emetine+U0126-treated oocytes as shown on (B). Statistical comparison was done using a two-tailed *t* test. **(E)** Mos–MAPK specifically downregulates PCM proteins, reduces the growth and number of astral and interpolar microtubules, and prevents anaphase B, thereby minimizing the size of the polar body. Bottom: schematic illustration of correlated changes in aster size and pole distance. Mean values shown on Fig. 4, B and C are plotted on a schematic of the spindle.

**Live-cell microscopy**

Starfish oocytes were imaged directly in the injection chambers. Live-cell images were acquired either on a Visitron Systems spinning disk confocal microscope or on a Zeiss LSM780 laser scanning confocal microscope. The spinning disk system is based on a Nikon Ti2 microscope body equipped with a Yokogawa W1 scanhead and two Prime BSI cameras (Teledyne Photometrics). On the laser scanning microscope, we used a

C-Apochromat 40×1.20 NA objective lens with the following settings: imaging area 100 × 100 µm, pixel size 200 nm, 15 z-slices with 2 µm step size, pinhole set to 2 Airy unit, and time intervals 3–5 min. On the spinning disk microscope, we used a CFI SR Plan Apo IR 60×1.27 NA water immersion objective. Typical settings were exposure time 200 ms, pixel size 110 nm, 11–15 z-slices with 3 or 4 µm step size, and time intervals 3–5 min. Images of the control and experimental groups were acquired under identical imaging conditions on the same microscope. All live imaging experiments were carried out at 20°C.

## Immunostaining
Oocytes were fixed at the desired times in a PFA/GA fixative (100 mM Hepes [pH 7.0], 50 mM EGTA, 10 mM MgSO₄, 0.5% Triton-X100, 400 mM sucrose, 2% formaldehyde, and in some experiments 0.2% glutaraldehyde) modified by Strickland et al. (2004). Subsequently, samples were permeabilized and blocked in blocking buffer (PBS + 0.1% Triton-X100 plus 3% BSA and the Image-IT reagent [Thermo Fisher Scientific]). Antibody staining was done for 24 h in a blocking buffer at room temperature with the primary antibody premixed with secondary nanobodies/Fab fragments. Microtubules were labeled with anti–α-tubulin mouse antibody (DM1A; Sigma-Aldrich), and lamin was labeled with a custom-made rabbit polyclonal antibody (Wesolowska et al., 2020) primarily labeled with Alexa 488 NHS ester (Thermo Fisher Scientific) according to the manufacturer's instructions. Secondary staining was done by anti-mouse secondary nanobodies with Atto-488 (#N2002-At488-S; NanoTag) or donkey Fab anti-mouse IgG AlexaFluor594 (#715-587-003; Jackson). DNA was labeled with Hoechst 33342 (Thermo Fisher Scientific). Oocytes were mounted in PBS or in the antifade agent ProLong Glass or SlowFade Diamond (Thermo Fisher Scientific). Fixed oocytes were imaged on the same microscope systems as used for live imaging with the following modifications: for the spinning disk confocal microscope, exposure time of 200 ms, pixel size of 110 nm, and 70 z-slices with 200 or 600 nm step size. On the laser scanning system, we used a pixel size of 40 nm, 80 z-slices with a 200 nm step size, and a pinhole set to 1 Airy unit.

## Image analysis
Prior to morphometric quantification, immunofluorescent images were deconvolved using the Huygens software (Scientific Volume Imaging) with either confocal, AiryScan, or spinning disk settings as appropriate. Quantification was done in ImageJ/FIJI (Schindelin et al., 2012). Images were rotated and resliced in XY and XZ planes to orient the spindle orthogonally. The quantification of spindle length and diameter of centrosomes was done by measuring the half-maximal width of the intensity profile along the respective structures along the spindle axis. Statistical analysis was performed by Excel (Microsoft); results were plotted in R (R Core Team).

The maximal length of microtubules and spindle diameter were quantified on live images using ImageJ/FIJI (Schindelin et al., 2012) in a maximal Z-projected volume slab of 10 µm thickness. The maximal length of microtubules was estimated by measuring the distance of aster tips to the centriole (live labeled

by EB3-mClover or by anti-tubulin staining in fixed samples), and the pole distance was measured by placing points at each pole and calculating their 3D distance. Statistical analysis was performed by using Excel (Microsoft); results were plotted in R (R Core Team).

For presenting images on figures, we typically used a Gaussian blur of 0.5–1 pixels to reduce shot noise. Images were adjusted for brightness and contrast, and where indicated, gamma adjustments were applied.

## Western blots
Approximately 50 oocytes were collected for each condition at the stated time points, lysed in SDS sample loading buffer (62.5 mM Tris, pH 6.8, 1.5% SDS, 0.5 M sucrose, 25 mM DTT, and 0.002% bromophenol blue), and heated at 95°C for 10 min. Proteins were resolved using SDS-PAGE performed on 4–12% Bis-Tris gels (Thermo Fisher Scientific) and transferred to a nitrocellulose membrane (#88025; Thermo Fisher Scientific) for western blotting. Anti–phospho-p44/42 MAPK (Erk1/2) antibody (#9101; Cell Signaling Technology) was diluted to 1:3,000, anti–phospho-cdc2 (Tyr15) antibody (#9111; Cell Signaling Technology) to 1:1,000, and anti-alpha tubulin (DM1A) to 1:1,000 in 5% BSA made in TBS with 0.1% Tween-20 (TBST). Fluorescently labeled anti-mouse CF770 (#20077; Biotium)/anti-mouse CF680 (#20065; Biotium) and anti-rabbit CF680 (#20418; Biotium)/anti-rabbit CF770 (#20484; Biotium) secondary antibodies were diluted to 1:10,000 in TBST before use. All immunoblots were scanned on a LI-COR Odyssey imager.

## Sample preparation and LC/MS analysis
Two or three biological replicates of oocytes were collected from different starfish animals for the proteomics and phosphoproteomics experiments, respectively (see schematics on Fig. 2, A and E). Oocytes were matured by the addition of 1-MA and were simultaneously treated with 20 µM U0126 or an equal amount of DMSO. Maturation timing was tested in prior experiments and was also directly monitored under a dissecting microscope. Synchronicity of oocytes was confirmed by staining a small portion of the collected oocytes with Hoechst 33342 and imaging on a confocal microscope. We confirmed a synchronicity of 92–96% of all samples analyzed.

For the phosphoproteomics experiment, 20 µl of starfish oocytes were lysed on ice in SDS lysis buffer (4% [wt/vol] SDS, 150 mM Hepes-KOH, pH 7.0, 2 mM DTT, 10 mM EDTA, 250 mM sucrose, and protease inhibitor cocktail [Complete Roche Mini]). The mechanical cell disruption was done by centrifuging oocytes through a 40-µm mesh at 7,000 g. Samples were diluted to a final 0.1% SDS, and 1 mM MgCl₂ was added. Proteins were reduced with 5 mM DTT for 30 min at 37°C, 300 rpm, and alkylated with 10 mM iodoacetamide (IAA) for 30 min at 25°C, 300 rpm, in the dark. The reaction was quenched with 10 mM DTT for 5 min at 25°C, 300 rpm.

Sample cleanup was performed by applying a single-pot, solid-phase–enhanced sample preparation protocol (Hughes et al., 2019). Briefly, carboxylate-modified magnetic beads (65152105050350, 45152105050250; Cytiva, mixed 1:1) were added at a 1:10 protein-to-bead mass ratio with an equal volume

of 100% (vol/vol) acetonitrile (ACN). Beads were washed twice with 80% (vol/vol) EtOH and once with 100% (vol/vol) ACN. Proteins were digested in 50 mM TEAB containing trypsin (Promega) at a 1:20 enzyme-to-protein ratio overnight at 37°C, 1,000 rpm. Peptides were removed from the magnetic beads, and aliquots were taken for whole-proteome analysis.

Residual samples were subjected to phosphopeptide enrichment according to the EasyPhos protocol (Humphrey et al., 2015). Briefly, peptide mixtures were adjusted to 228 mM KCl, 3.9 mM $KH_2PO_4$, 38% (vol/vol) ACN, and 4.5% (vol/vol) trifluoroacetic acid (TFA), mixed, and cleared by centrifugation. $TiO_2$ beads (10 µm, GL Sciences) in 80% ACN and 6% TFA were added at a beads-to-protein ratio of 10:1. Peptides were incubated with beads for 20 min at 40°C, 1,000 rpm. Beads were resuspended and washed in 60% (vol/vol) ACN and 1% (vol/vol) TFA four times. Finally, beads were resuspended in 80% (vol/vol) ACN and 0.5% (vol/vol) acetic acid and transferred to empty columns (74-3840; Harvard Apparatus). Phosphopeptides were eluted in two steps with 40% (vol/vol) ACN and 15% (vol/vol) $NH_4OH$.

TMTsixplex (Thermo Fisher Scientific) labeling was performed according to the manufacturer's instructions. Whole-proteome and phosphopeptide-enriched samples were labeled separately. Both were further cleaned using C18 MicroSpin columns (Harvard Apparatus) according to the manufacturer's instructions. Samples were dissolved in 10 mM $NH_4OH$, pH 10, 5% (vol/vol) ACN and peptides were loaded onto an XBridge C18 column (Waters) using an Agilent 1100 series chromatography system. The column was operated at a flow rate of 60 µl/min with a buffer system of buffer A, 10 mM $NH_4OH$ pH 10; and buffer B, 10 mM $NH_4OH$ pH 10, 80% (vol/vol) ACN. The column was equilibrated with 5% B and separation was performed over 64 min using the following gradient: 5% B (0–7 min), 8–30% B (8–42 min), 30–50% B (43–50 min), 90–95% B (51–56 min), and 5% B (57–64 min). For phosphopeptides, the first 6 min were collected as flow-through, followed by 48 × 1 min fractions, which were reduced to 23 fractions by concatenated pooling. The remaining fractions were discarded. For the whole-proteome sample, 54 × 1-min fractions were collected and combined into 19 final fractions. The remaining fractions were discarded. Fractionated whole-proteome and phosphopeptide samples were dried in a speed vac concentrator and subjected to LC-MS/MS analysis.

For the proteomics time course experiment, oocytes were collected, pelleted, and lysed in SDS lysis buffer (4% [wt/vol] SDS, 150 mM Hepes-NaOH, pH 7.5, and 1× Roche Protease inhibitor cocktail) by incubation for 5 min at 99°C; by sonication, in alternating 30 s on- and 30 s off-cycles at highest intensity output (Bioruptor, Diagenode); and by another incubation for 5 min at 99°C. Protein concentrations were determined using Pierce BCA Protein Assay Kit - Reducing Agent Compatible (23250; Thermo Fisher Scientific). Proteins were reduced with 10 mM DTT for 30 min at 37°C, 300 rpm, and alkylated with 40 mM IAA for 30 min at 25°C, 300 rpm, in the dark. The reaction was quenched with 10 mM DTT for 5 min at 25°C, 300 rpm. Aliquots containing 0.1 mg of protein for each replicate of each sample were further processed. Samples were diluted to a final 0.4% (wt/vol) SDS, and 1 mM $MgCl_2$ was added. DNA content was digested using 312.5 U of Pierce Universal Nuclease

(Thermo Fisher Scientific) for 2 h at 37°C, 300 rpm. Samples were cleaned up by single-pot, solid-phase–enhanced sample preparation protocol as above. Peptides were recovered from the magnetic beads, and the beads were washed once with 100 µl water. Samples were dried in a speed vac. Samples were measured using DIA.

**Proteomics data analysis**
The proteome database was compiled using datasets at Echinobase v2 (https://echinobase.org) (Kudtarkar and Cameron, 2017) and NCBI annotation (RefSeq ID 23626818) for *P. miniata*. The available protein sequences were clustered with the mmseqs2 easy-clust workflow (Steinegger and Söding, 2018) with a sequence similarity threshold of 0.95 to obtain a nonredundant set of representative sequences. Transcript sequences for Echinobase v2 were derived from the genome sequence and transcriptome annotation with the gffread tool (Pertea and Pertea, 2020). The representative transcripts were annotated via the TRAPID web interface (Bucchini et al., 2021) using the EggNOG 4.5.1 database (Huerta-Cepas et al., 2016). The full log of TRAPID annotation is deposited at https://doi.org/10.25625/03K7BQ. In addition, manually curated protein sequences for selected proteins of interest were included in the proteome. TRAPID gene identifiers and ontology terms were used for the downstream overrepresentation analysis.

Raw phospho-TMT mass spectrometry data were processed using MaxQuant version 1.6.17.0 (Cox and Mann, 2008). MaxQuant output tables were filtered to remove decoys, contaminants, and low-confidence peptides with posterior error probability ≥0.01. Ratios of phosphopeptide intensities to the corrected parent protein intensities were $\log_2$-transformed and used for subsequent differential expression analysis. The protein groups were not post-filtered by identified unique peptides or q values to maximize the match with phosphopeptides, resulting in 76% of phosphopeptides having a corresponding total protein intensity for normalization.

A linear model testing for the difference between means of phosphopeptide ratios in treatment and control groups was fit with least squares, and P values were estimated with the empirical Bayes approach via the limma R package (Ritchie et al., 2015). Multiple hypothesis correction was done using the Benjamini and Hochberg approach for false discovery rate estimation, and a threshold of P-adjusted ≤0.05 was set (Benjamini and Hochberg, 1995).

GO overrepresentation analysis was performed using the g:Profiler tool g:GOSt using the default P value adjustment strategy "g_SCS" and a significance threshold of 0.05 (annotation available with identifier "gp__EGla_34PO_4UY") (Raudvere et al., 2019).

The heatmap of all significantly differentially phosphorylated sites was made from the normalized ratios of significantly up- and downregulated phosphopeptides scaled to the mean of 0 and a SD of 1. Plotting was done via the pheatmap R package with the "ward.D2" clustering method (Murtagh and Legendre, 2014) and Euclidean distance as a metric (Kolde, 2025).

The phospho (STY) probabilities for each tested peptide sequence were mean aggregated on the amino acid level across all peptide-spectrum matches (PSMs) and the most likely amino

acid per peptide sequence was treated as phosphorylated. Peptide sequences, log$_2$-fold changes, and P values of differential expression were submitted for kinase prediction at https://phosphosite.org (Hornbeck et al., 2015). The top significantly downregulated kinases that passed the p-adjusted threshold of 0.05 were reported. Phosphosite motifs were derived by centering all the differentially dephosphorylated peptides (q value <0.05) around the most likely phosphosites as a foreground, compared with the background of all dephosphorylated peptides, and plotted via the Logolas R package (Dey et al., 2018). Kinase-specific motifs were derived from a set of significantly dephosphorylated peptides having the kinase of interest in the top 10 predicted targets.

DIA proteomics was processed with DIA-NN using the default settings (Demichev et al., 2020), and the signal intensity was post-aggregated with the MaxLFQ algorithm upon the following PSM-level filters: Q value ≤ 0.001 and protein group Q value ≤ 0.05. Same as for phospho-TMT, the differential expression was assessed via "limma," with additional weights of 0.05 assigned to the missing values, conservatively imputed by minimal replicate intensity. The thresholds of adjusted P value (BH procedure) of 0.05 and a log$_2$FoldChange of 0.5 were chosen to define reliably quantified proteins with a large enough effect size.

### Statistical analyses

Statistical comparisons were done using the Mann–Whitney test or a two-tailed $t$ test, as indicated in the figure legend. ** denotes $P < 0.05$, and *** denotes $P < 0.0001$. Data distribution was assumed to be normal, but this was not formally tested. Computational and statistical analysis has been implemented in R v4.1.1 (Ihaka and Gentleman, 1996) or in Excel (Microsoft) and subsequently plotted in R.

### Online supplemental material

This manuscript contains 4 supplementary figures/tables and 3 supplemental data files. Fig. S1 shows experiments to determine the effective dose of U0126 and morphometric analyses of spindles treated with U0126. Fig. S2 shows further analyses of the proteomics experiments. Fig S3 shows the effect of preventing cyclin B synthesis during meiosis. Table S1 shows tabulated data for the functional modules shown in Fig. 3 B. The supplemental data files include the complete phospho-proteomics dataset (Data S1), the complete proteomics time course dataset (Data S2), the significant hits of the phospho-proteomics experiment (Data S3), and the uncropped membranes of the western blots shown (Source Data F1 and SourceData FS1).

### Data availability

No new or unique reagents were generated in this study. The TRAPID protein database used for searching the proteomics data is available at GRO.data at https://doi.org/10.25625/03K7BQ. The mass spectrometry proteomics datasets for starfish oocytes are available in PRIDE (Perez-Riverol et al., 2022) and UCSD MassIVE member repositories of the ProteomeXchange Consortium with the project accession identifiers PXD043650 and PXD067579|MSV000098905, respectively.

## Acknowledgments

We thank all members of the Lenart laboratory for protocols, reagents, and support, in particular Jasmin Jakobi and Antonio Politi (MPI-NAT, Göttingen, Germany). We thank MPI-NAT's Animal Facility, specifically Sascha Krause and Ulrike Teichmann. We would like to thank So (Nick) Chun and Melina Schuh (MPI-NAT, Göttingen, Germany) for providing the recombinant EB3-mClover3 protein, Pim J. Huis In't Veld and Andrea Musacchio for Cdk1–cyclin B complex (MPI for Molecular Physiology, Dortmund, Germany), and Dirk Görlich (MPI-NAT, Göttingen, Germany) for the Gibson assembly kit.

Ivan Avilov is a member and has been partially funded by the IMPRS Molecular Biology Program. The laboratories of Peter Lenart, Juliane Liepe and Henning Urlaub are funded by the Max Planck Society. Peter Lenart receives additional funding from the Deutsche Forschungsgemeinschaft through the collaborative grant LE 2926 (project number 505673695), and the Research Training Group 'CYTAC' (RTG 2756 project B4, project number 449750155). Henning Urlaub receives additional funding from the Deutsche Forschungsgemeinschaft through the Collaborative Research Centre SFB1565 (project number 469281184). Open Access funding provided by the Max Planck Society.

Author contributions: Ivan Avilov: conceptualization, data curation, formal analysis, investigation, methodology, project administration, and writing—original draft, review, and editing. Yehor Horokhovskyi: conceptualization, data curation, formal analysis, software, visualization, and writing—original draft, review, and editing. Pooja Mehta: investigation and writing—review and editing. Luisa Welp: data curation and investigation. Jasmin Jakobi: investigation. Mingfang Cai: formal analysis, investigation, methodology, and visualization. Aleksander Orzechowski: investigation, resources, and validation. Henning Urlaub: funding acquisition, methodology, supervision, and writing—review and editing. Juliane Liepe: formal analysis, supervision, and writing—review and editing. Peter Lenart: conceptualization, data curation, formal analysis, funding acquisition, investigation, methodology, project administration, resources, supervision, validation, visualization, and writing—original draft, review, and editing.

Disclosures: The authors declare no competing interests exist.

Submitted: 2 January 2024

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

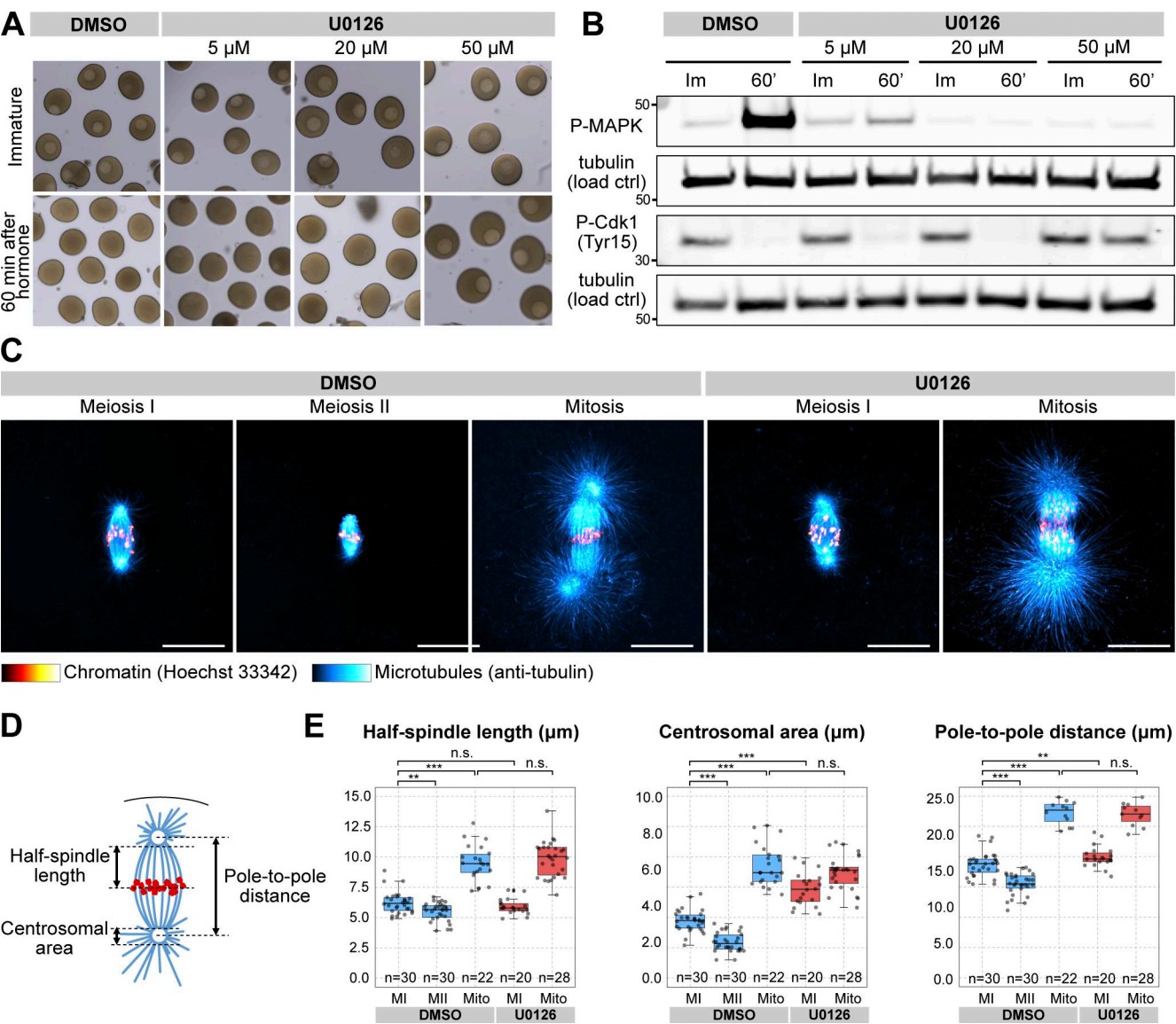

Figure S1. **U0126 effectively and specifically inactivates Mos–MAPK in starfish oocytes and alters spindle morphology. (A)** Oocytes were treated with the indicated concentrations of U0126 concomitant with maturation hormone addition. Thereafter, maturation was monitored by videomicroscopy, of which selected frames are shown at the start of the recording and 60 min after hormone addition. **(B)** Western blots of oocytes treated as in A using phospho-MAPK and phospho-Cdk1 antibodies to monitor Mos–MAPK and Cdk1 activities, respectively. Numbers to the left are in kDa and mark the position of the molecular weight markers. For the complete blots, see Source Data FS1. **(C)** Immunofluorescence images of oocytes stained for microtubules (anti-tubulin, cyan) and chromatin (Hoechst 33342, red) at the indicated stages and treated with 20 µM U0126 or an equal amount of DMSO. Imaging was done on a laser scanning confocal microscope. Maximum intensity projections are shown; scale bars are 20 µm. **(D)** Schematics of the meiotic spindle illustrating the morphometric measurements. Microtubules are cyan, and chromatin is red. **(E)** Individual data points and boxplots showing 3D measurements performed on images similar to those shown in C. **P < 0.05, ***P < 0.0001; statistical analysis was done using a two-tailed t test. n is the number of oocytes analyzed. Source data are available for this figure: SourceData FS1.

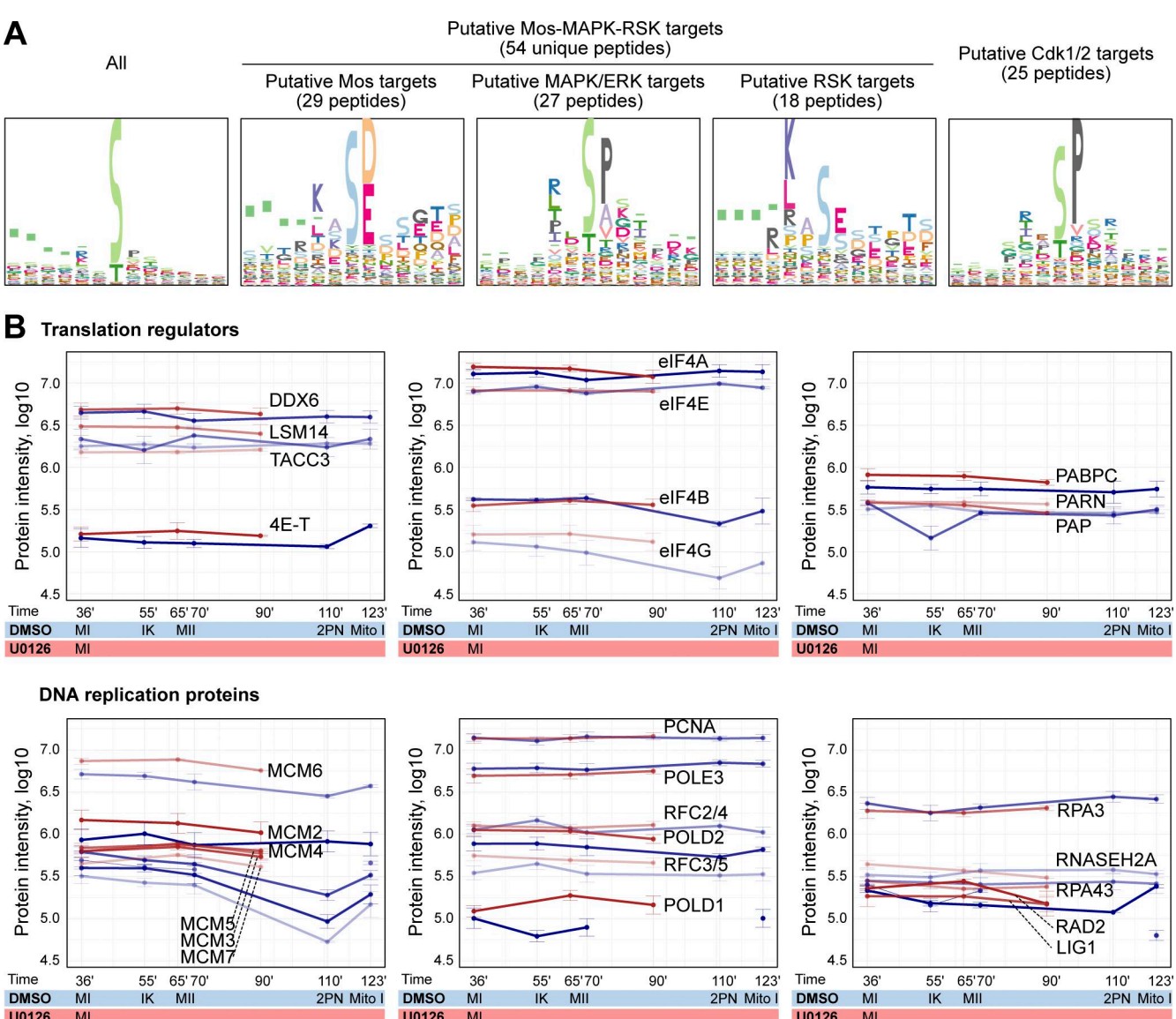

Figure S2. **Further analyses of the proteomics experiments. (A)** Logo motif representation of the consensus site of all differentially phosphorylated peptides in the phosphoproteomics experiment. Peptides were grouped based on the kinases that would be most likely to be responsible for their phosphorylation. Logo motif representation of the consensus sequence for each group most likely to be phosphorylated by Mos, MAPK, RSK, and Cdk1/2, respectively. The y-axis shows the information content that was scaled between 0 and 1 for each of the kinase targets. **(B)** $Log_{10}$ normalized protein intensities for the indicated proteins in the proteomics time course experiment. The mean and standard error are shown of the two biological replicates and three technical repetitions of each.

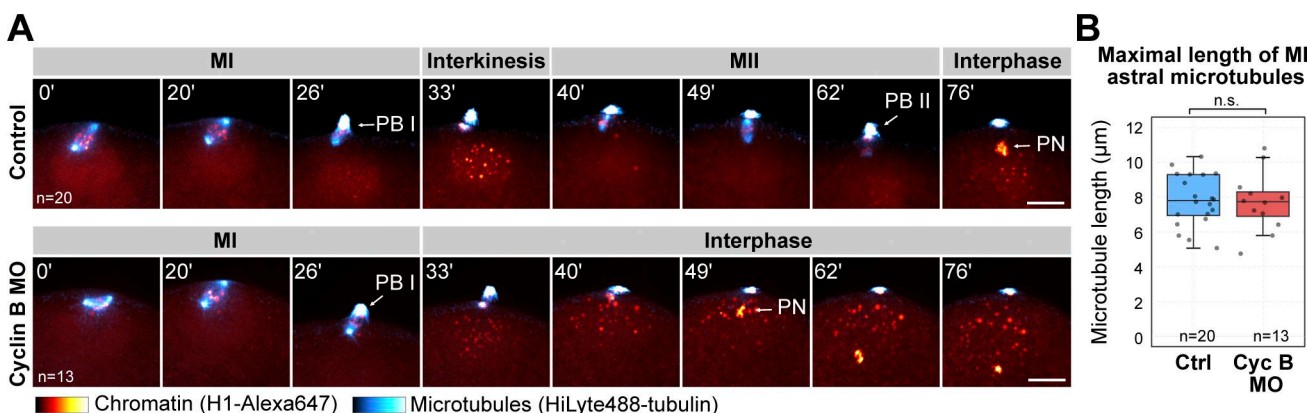

Chromatin (H1-Alexa647)   Microtubules (HiLyte488-tubulin)

Figure S3.   **Preventing cyclin B synthesis blocks MII transition, but it does not affect MI spindle morphology. (A)** Selected frames from a time-lapse recording of oocytes injected with H1-Alexa647 (red) and HiLyte488-tubulin (cyan) to label chromosomes and microtubules, respectively. Oocytes were imaged through meiosis on a spinning disk confocal microscope. Before inducing maturation, oocytes were injected with either a morpholino targeting cyclin B or water as control. Maximum intensity projections are shown, scale bars are 20 µm, and time is given in minutes. PB I and II denote the first and second polar bodies, respectively; PN is the pronucleus. **(B)** Individual data points and boxplots showing a comparison of the maximal length of microtubules at metaphase I between control and cyclin B morpholino-injected oocytes. Statistical comparison was done using a two-tailed *t* test; *n* is the number of oocytes analyzed.

Provided online are Table S1, Data S1, Data S2, and Data S3. Table S1 shows tabulated data for the functional modules shown in Fig. 3 B. Data S1 shows the phosphoproteomics dataset (DDA-TMT). Data S2 shows the proteomics time course experiment (DIA). Data S3 shows significant hits of the phosphoproteomics experiment.

