## [Peer Review File · The Journal of Cell Biology]

Phosphoproteomic identification of Mos-MAPK targets in meiotic cell cycle and asymmetric division

Ivan Avilov, Yehor Horokhovskiy, Pooja Mehta, Luisa Welp, Jasmin Jakobi, Mingfang Cai, Aleksander Orzechowski, Henning Urlaub, Juliane Liepe, and Peter Lenart

Corresponding Author(s): Peter Lenart, Max Planck Institute for Multidisciplinary Sciences

Review Timeline:

Submission Date:	2024-01-02
Editorial Decision:	2024-02-09
Revision Received:	2024-12-23
Editorial Decision:	2025-03-17
Revision Received:	2025-08-07
Editorial Decision:	2025-08-07
Revision Received:	2025-08-28

Monitoring Editor: William Bement

Scientific Editor: Dan Simon

Transaction Report:

DOI: <https://doi.org/10.1083/jcb.202312140>

February 9, 2024

Re: JCB manuscript #202312140

Dr. Peter Lenart
Max Planck Institute for Multidisciplinary Sciences
Research Group Cytoskeletal Dynamics in Oocytes
Am Fassberg 11
Göttingen 37077
Germany

Dear Dr. Lenart,

Thank you for submitting your manuscript entitled "Phosphoproteomics identifies targets of Mos-MAPK regulating translation and the spindle in oocytes." The manuscript has been assessed by expert reviewers, whose comments are appended below. Although the reviewers express potential interest in this work, significant concerns unfortunately preclude publication of the current version of the manuscript in JCB.

You will see that the reviewers feel that your work addresses an important topic and presents interesting data. However, they also state that additional controls are required in order to confirm that the observed effects in U0126-treated cells are specifically due to Mos-MAPK inhibition and new experiments are needed to investigate in more detail how Mos-MAPK phosphorylation targets regulate translation in oocytes. Additionally, they ask you to reconcile differences in your results with prior papers as well as note several references that should be discussed and provided other comments aimed at clarifying and better explaining the data and conclusions which you will need to address by text and figure revisions.

Please let us know if you are able to address the major issues outlined above and wish to submit a revised manuscript to JCB. Note that a substantial amount of additional experimental data likely would be needed to satisfactorily address the concerns of the reviewers. The typical timeframe for revisions is three to four months. While most universities and institutes have reopened labs and allowed researchers to begin working at nearly pre-pandemic levels, we at JCB realize that the lingering effects of the COVID-19 pandemic may still be impacting some aspects of your work, including the acquisition of equipment and reagents. Therefore, if you anticipate any difficulties in meeting this aforementioned revision time limit, please contact us and we can work with you to find an appropriate time frame for resubmission. Please note that papers are generally considered through only one revision cycle, so any revised manuscript will likely be either accepted or rejected.

If you choose to revise and resubmit your manuscript, please also attend to the following editorial points. Please direct any editorial questions to the journal office.

GENERAL GUIDELINES:

Text limits: Character count is < 40,000, not including spaces. Count includes title page, abstract, introduction, results, discussion, and acknowledgments. Count does not include materials and methods, figure legends, references, tables, or supplemental legends.

Figures: Your manuscript may have up to 10 main text figures. To avoid delays in production, figures must be prepared according to the policies outlined in our Instructions to Authors, under Data Presentation, <https://jcb.rupress.org/site/misc/ifora.xhtml>. All figures in accepted manuscripts will be screened prior to publication.

Supplemental information: There are strict limits on the allowable amount of supplemental data. Your manuscript may have up to 5 supplemental figures. Up to 10 supplemental videos or flash animations are allowed. A summary of all supplemental material should appear at the end of the Materials and methods section.

Please note that JCB now requires authors to submit Source Data used to generate figures containing gels and Western blots with all revised manuscripts. This Source Data consists of fully uncropped and unprocessed images for each gel/blot displayed in the main and supplemental figures. If your revised manuscript will include cropped gel and/or blot images, please be sure to provide one Source Data file for each figure that contains gels and/or blots along with your revised manuscript files. File names for Source Data figures should be alphanumeric without any spaces or special characters (i.e., SourceDataF#, where F# refers to the associated main figure number or SourceDataFS# for those associated with Supplementary figures). The lanes of the gels/blots should be labeled as they are in the associated figure, the place where cropping was applied should be marked (with

a box), and molecular weight/size standards should be labeled wherever possible. Source Data files will be made available to reviewers during evaluation of revised manuscripts and, if your paper is eventually published in JCB, the files will be directly linked to specific figures in the published article.

If you choose to resubmit, please include a cover letter addressing the reviewers' comments point by point. Please also highlight all changes in the text of the manuscript.

Regardless of how you choose to proceed, we hope that the comments below will prove constructive as your work progresses. We would be happy to discuss them further once you've had a chance to consider the points raised. You can contact the journal office with any questions at cellbio@rockefeller.edu.

Thank you for thinking of JCB as an appropriate place to publish your work.

Sincerely,

William Bement, PhD
Monitoring Editor
Journal of Cell Biology

Dan Simon, PhD
Scientific Editor
Journal of Cell Biology

Reviewer #1 (Comments to the Authors (Required)):

In the present study Avilov et al perform a global phosphoproteome analysis of starfish oocytes in metaphase of meiosis I, in control and UO126-treated oocytes. They find that inhibition of the Mos-MAPK pathway affects the meiotic nature of the spindle and Cyclin B translation to drive the second meiotic division. They also found that entry into meiosis I is not affected by UO126, and that fertilization arrest is abolished, however, this has been published previously.

Main points:

The experiments are based on previous results published in 2000 in starfish oocyte by Tachibana et al. In this paper, a Mos antisense RNA approach was used, which is much more specific than UO126 to block the Mos-MAPK pathway. Rescue experiments with recombinant GST-Mos were used as controls. Without Mos, 3 cell divisions and DNA replication right after metaphase I were observed most of the time. MAPK activation using phospho-MAPK was analysed and Cyclin A and Cyclin B expression levels were assessed by western blots. The authors need to add these controls (western blots to assess Cyclin B levels, replication immediately after metaphase I, MAPK activity) to make sure that the effect observed is indeed due to Mos-MAPK inhibition alone.

I have some concerns regarding the UO126 concentrations used (20 μ M final). In *Xenopus* oocytes, 50 μ M of UO126 is able to inhibit other kinases than MAPK (monitored by MBP kinase assays) (Dupré et al., 2002, *Embo J*). Can the authors exclude that 20 μ M of UO126 do not show the same unspecific effect in starfish oocytes? More importantly, and still in *Xenopus*, UO126 decreases Cdk1 activity at GVBD because the Mos-MAPK likely acts as a positive regulator of Cdk1 activation (Gross et al, 2000, *Current Biology*; Dupré et al., 2022, *Embo J*). First, this is not observed when oocytes are injected with Mos antisense RNAs to target the Mos-MAPK pathway, and second, it has to be determined whether the same is happening in starfish oocytes. In short, the authors need to determine the concentrations that only affect Mos-MAPK and not Cdk1 at the experimental conditions they are using.

Cdk1 and MAPK phosphorylate redundant "SP" sites. The authors performed a phosphoproteome to identify proteins phosphorylated by the Mos-MAPK pathway. These proteins are less phosphorylated in UO126-treated oocytes than in WT oocytes. Since Cdk1 activity may be decreased in UO126, phosphorylation of proteins may be affected by Cdk1 and not MAPK inhibition. For these reasons, Cdk1 activity must be absolutely assayed in metaphase I to make sure that the identified proteins are targeted specifically by the Mos-MAPK pathway.

Related to this point, have the author checked the kinase consensus motif of the dephosphorylated phosphosites? Direct Mos-MAPK targets should be SP/TP phosphosites. Since Cdk activity is expected to be affected by UO126 treatment as well, a second check is that the targets are not Cdk sites. Even if the kinase consensus site of Cdk and Mos-MAPK overlap (SP/TP),

Cdk precisely recognises SPx(K/R) and TPx(K/R), which is a first way to discriminate between Cdk and Mos-MAPK targets. The author should provide a logo motif analysis, which is a first quick way to visualize the data.

The statement in the title saying the Mos-MAPK pathway regulates translation is an overinterpretation. There are no data assessing protein translation globally, and in the paper by Tachibana et al there was no effect on Cyclin A and B protein levels when Mos translation was blocked. This discrepancy has to be addressed. Furthermore, in mouse and xenopus oocytes, CPEB phosphorylation promotes the degradation of the protein in metaphase I. What is happening in starfish oocytes is unknown. A western blot may be necessary to make sure that CPEB is still present. They also mentioned that they did a proteomic experiment, in which they do not observe significant changes in protein levels. The data are not shown and a supplementary figure with these data should be included.

The article uses various techniques to show that U0126-treated cells enter mitosis after meiosis I. In Figure 1, the decondensation of chromosomes mentioned by the authors is not visible at this resolution. However, the reformation of the lamina is a good argument, but a control is missing showing that in control starfish oocytes there is no lamina in anaphase I, 60 min after NEBD. In Figure 1, the authors also provide a very detailed description of spindle morphology, which is convincing. In Figure 4, a different technique is used - looking at the formation of phase-separated condensates of H1-Alexa647 in the context of MO Cyclin B and emetin treatment. Why didn't the authors use the same controls as in Figure 1 for consistency? The lamina should also have been shown. And far more importantly, the author argues that "By contrast, inhibition of Cyclin B translation by morpholino (Fig 4A, B) or by general translation inhibition (Fig. 4D) did not affect the morphology of the MI spindle. This indicates that Mos-MAPK affects MI spindle morphology, and it does so independently of translational regulation". These data are not shown and an analysis similar to Figure 1F is required to draw this conclusion must be provided. Furthermore, this result is very important to demonstrate that the phenotypes are independent of Cyclin B1-Cdk and specific to Mos-MAPK. It seems that there are lots of proteins identified that are involved in DNA replication, when comparing untreated and UO126-treated oocytes. These proteins should be listed in Figure 3B as it may help understanding how Mos-MAPK activity may contribute to inhibition of DNA replication between meiosis I and meiosis II. Can the authors assess protein levels of key proteins involved in replication?

Minor comments:

Regarding the mass spectrometry data and Figure 2, please include the following comments in the MS analysis part in the methods, text and/or figure legends:

- Figure 2A: When were the oocytes treated and for how long before sample collection? the authors also mentioned that they tested cell cycle synchrony (data should be shown in Supp).
- Figure 2B: Numbers in the figure do not match those in the text.
- Figure 3C: How many peptides are shown in this hierarchical clustering? Only the 269 high fold change peptides? There are also more than 300 phosphosites shown in the supplementary data. A supplemental dataset with a list of these unique peptides would be appreciated.
- Figure 3D and 3E: missing description in Materials and Methods

In general, I was surprised that almost all phosphosites show a decrease in phosphorylation. Only a very small percentage looks phosphorylated. Surely many other substrates including kinases and phosphatases are under control of Mos-MAPK. I would have expected to see a lot of non-direct phosphorylations as well, especially if the U0126 treatment is longer than 5-10 min (this allows enough time for other kinase activation or phosphatase inactivation). Can the author comment on this point?

Throughout the figure and the text, please state whether the oocytes are in metaphase I or II and not in metaphase "alone" (the same applies to the other stages). This applies particularly to Figures 5 and 6.

Reviewer #2 (Comments to the Authors (Required)):

Using starfish oocytes as model system, the authors address the function of Mos kinase in oocyte maturation. Mos-MAPK functions are conserved in starfish involving reactivation of CDK1/Cyclin B at the end of MI and a CSF-like arrest, that occurs at the G1 phase. Using U0126, a MEK kinase inhibitor, the authors reveal that loss of the Mos/MAPK pathway cause oocytes to proceed directly from MI to parthenogenetic development. To understand the underlying mechanisms, the authors then perform phospho-proteomics analyses of control and U0126-treated starfish animals. These analyses resulted in the identification of 269 unique phospho-peptides passing stringent filtering criteria. Affected processes were translation, cell cycle and cytoskeleton regulation. Among the translational regulators were CPEB and eIF4ENIF1 and therefore the authors conclude that Mos-MAPK might induce CPE-dependent translation. Specifically, the hypothesize that Mos-MAPK may drive MII by inducing Cyclin B translation. As expected, oocytes not expressing Cyclin B were able to complete MI, but failed to enter MII. Conversely, ectopic CDK1/Cyclin B rescued the failure of U0126-treated oocytes to enter MII. Inhibition of Cyclin B synthesis did not affect the morphology of MI spindles indicating that Mos/MAPK affect MI spindles independently of translational regulation. In fact, the authors find evidence that Mos/MAPK downregulates centrosomal MT nucleation and thereby contributes to minimizing PB size. In the absence of Mos activity, the MI spindle becomes mitotic-like with astral MTs pushing the spindle away from the cortex .

This manuscript addresses an important aspect of meiosis, i.e., what are the targets of Mos/MAPK in oocytes. As was to be expected from Peter Lenart's group, the experiments were carried out very thoroughly and the data presented are of very high quality. The manuscript will be frequently cited in the future because the role of Mos/MAPK is still unclear. This reviewer thinks that the part on MT dynamics is more convincing than the part on translational regulation, see below for details. This reviewer therefore recommends publication of the paper once the authors have addressed the following points.

The main concern of this reviewers relates to the translational regulation part:

- Fig. 4: I think it would be very helpful to see the protein levels of cyclin-B in Control and cyclin-B MO-injected oocytes over the course of meiotic maturation (if there is an antibody available).
- I think it is also not surprising that you need cyclin B synthesis for entry into MII. This has been shown in many species and I think describing the link to the Mos-MAPK pathway is more interesting. However, in my opinion, from the data presented here it is not possible to say if translation of cyclin B is directly affected by the Mos-MAPK pathway. Another possibility could be that cyclin B translation after MI works perfectly fine upon U0126, but Cdk1 is inactivated by inhibitory phosphorylation (Myt1 and Cdc25 are hits in the MS analysis) and the oocytes therefore end up in interphase. This would probably also be rescued by the ectopic Cdk1-CycB used in Fig. 4D (probably depending on how much Cdk1 activity was added compared to endogenous Cdk1 activity - is this known?). In my opinion, this figure would be much more convincing, if the authors could show that translation of a cycB 3'UTR reporter mRNA is affected by inactivation of the Mos-MAPK pathway. The effect could be still indirect, but it would provide more evidence that Mos/MAPK indeed regulates translation. The authors should discuss in more detail the possibility that the effects on translation upon U0126 treatment are indirect or provide data, e.g., in vitro assays, that the effects are direct.
- Furthermore, the authors write in M&M that they use the cyclin-B MOs at 500mM final concentration, which is much more than expected. Is this a typo?
- Page 5: the authors write „For maintenance of the CSF arrest, Mos-MAPK appears to act on Erp1/Emi2 to prevent Cyclin B degradation by the APC/C (Wu and Kornbluth, 2008). However, the detailed molecular mechanisms remain incompletely understood." There have been follow-up publications working on this mechanism and it has been shown, that p90Rsk phosphorylation of Erp1/Emi2 is required for efficient recruitment of PP2A-B56, which counteracts inhibitory/destabilizing phosphorylations (Isoda et al., 2011 and Hertz et al., 2016).
- Fig. 2B: in the text, the authors write "In total, we quantified 9,035 protein groups, of which 4,716 contained phospho-peptides, 22,822 phospho-peptides in total (Fig. 2B)." To me it is not clear, what the authors refer to as "protein groups"? How do they relate to the 8703 proteins of the left panel in Fig. 2B? It is also not entirely clear to me how the numbers of the left panel in Fig. 2B correlate to the numbers given in the text.
- Fig. 2: the authors describe that many translation regulators are differentially phosphorylated in the U0126-treated oocytes, suggesting that Mos/MAPK are required for translational control during maturation. However, in their MS analysis they found no differentially expressed proteins. Was this expected? In the reference cited (Swartz et al., 2021) there is a small group of proteins that increase in abundance during maturation (already in MI). Are these affected by the U0126 treatment? Later, cyclin B is proposed to be changing in abundance upon U0126 treatment. Are other CPEB-regulated mRNAs expected to be regulated by Mos-MAPK as well? Or is this mechanism in starfish meiosis not used as prominently as in other model organisms? The Mos mRNA was also described to be regulated by CPEB in other species. Do the authors know if the translational upregulation of Mos mRNA itself is regulated by the MAPK pathway in starfish oocytes?
- Page 10: The authors write that "In addition, we detected several changes indicative of an overall signature of M-phase exit, including regulators of DNA replication". Is this expected considering that the U0126-treated samples showed a 92-96% synchronicity in metaphase?
- Page 11 / Figure 3B: the authors write that it has been shown that eIF4ENIF1 and Maskin/TACC3 are part of the same complexes. To my knowledge this has never been shown (or suggested). Both proteins would also directly compete for eIF4E
- Fig. 5B: the authors write that they observe a "slightly increased pole-to-pole distance" in U0126-treated metaphase-I spindles. However, in Figure 1G the half-spindle length in MI was not increased. Although the effect in Fig. 5B is small, can the authors say why there is this discrepancy?
- Fig. 6C: Similarly, it is not clear to me why in Fig 6C, there is a clear increase in width of the metaphase spindle upon U0126 treatment, whereas there is no difference in MI spindle width in Fig. 1G
- Fig. 7B: It would help to indicate in the Figure what the conditions on the left and right side are
- Page 18: the authors write "transnational regulation"
- p14, bottom: "...twice in diameter in Mos-MAPK inhibited oocytes as compared to control (Fig. 5F)". It should be 5E.

- p15, first sentence: It should be 5F, not 5G
- p15: "..., which contain much more microtubules and reach a size of 20um towards..." Where are the data shown?
- p18: "...., these may use similar molecular mechanisms to maintain arrest by maintaining high levels of Cdk1-Cyclin B activity. This reviewer is puzzled. Does this statement also apply to starfish oocytes, which arrest in G1?

Reviewer #3 (Comments to the Authors (Required)):

During oogenesis the cell cycle arrests in prophase of meiosis I. Oocyte maturation is promoted by a hormonal signal that triggers the advance through the meiotic divisions. Oocyte activation triggers a series of changes resulting in the increased translation of key cell cycle regulators, such as cyclin B, that promote entry into MI. An important aspect of meiosis is that cells do not fully exit from MI but rapidly progress into MII to perform a reductive division. Extensive work in *Xenopus* and *Drosophila* has documented numerous changes in mRNA polyadenylation and protein translation that are important for progression through the meiotic divisions. In all systems studied CPEB is critical for control of mRNA polyadenylation and progression through meiosis. Avilov et al. describe their work to understand how the Mos kinase contributes to progression through the meiotic divisions. They use live and fixed cell imaging in wild-type and Mos-inhibited oocytes to determine that Mos is required after MI to promote progression into MII. This result is consistent with previous biochemical studies characterizing the timing of Mos activation in starfish oocytes. Additionally, the authors perform phosphoproteomics to identify phosphorylation sites that decrease after Mos inhibition. They identify a large number of altered phosphorylation sites and speculate about how these phosphorylation events may result in the observed cellular phenotypes. Overall, this study provides a thorough cell biological characterization of the Mos inhibition phenotype in starfish oocytes but does not provide any new insight into the critical substrates or molecular mechanisms. Additionally, the authors need to integrate their results with the extensive literature investigating changes in mRNA polyadenylation and translation during oocyte maturation.

Major Points

1. In Figures 2 and 3 the authors report a proteomic analysis of phosphorylation sites that increase and decrease after Mos inhibition. This data is a useful starting point to understand the molecular functions of Mos during meiosis. The major concern to this area of the manuscript is that the authors do not pursue these results further. The authors include speculation about the potential functions of these phosphorylation sites, but none are tested individually and none are validated using an orthogonal method. In order for this work to be an important contribution to the field it would be necessary to understand how some of the individual phosphorylation sites lead to the observed cellular phenotypes. Since there is a huge body of work examining mRNA polyadenylation during oocyte maturation (see below) that could be an interesting starting point.
2. There is a tremendous amount of literature about the molecular changes that mediate oocyte maturation in *Xenopus* and *Drosophila* that the authors have not cited or discussed in their work. For example, there are intricate feedback loops that govern CPEB-mediated mRNA polyadenylation and protein translation in *Xenopus* (PMID: 20531391, 18536713, 18385675, 18267074, 31896558) and extensive characterization of the same processes in *Drosophila* (27474798, 25349405, 24882012). Since a major claim of this manuscript is that the translation regulation module is the key to the MI to MII transition it is essential that the authors cite and discuss these studies.
3. The authors speculate that Mos directly phosphorylates CPEB to mediate cyclin B translation between MI and MII. However, this is in contrast to the observations in *Xenopus* and mouse oocytes where Aurora-A (Eg2) is responsible for CPEB phosphorylation (11526086, 11106762, 10749216). The authors need to cite these studies and discuss these previous results in the context of their observation that Aurora-A kinases are repressed after Mos inhibition.
4. In Figure 2E the authors describe families of kinases that exhibit lower activity after Mos inactivation. It would be useful to provide information about how many phosphosites are direct targets of Mos vs. indirectly regulated targets.

Minor Points

1. The plot in Figure 2F is completely unreadable. The authors should simplify this plot by removing the gene names and only including the names of a few key proteins that they would like to highlight.
2. The phosphopeptides present in Figure 3A are hardly discussed in the text and do not meaningfully contribute to the presentation of the data. This data would be more appropriate for a Supplemental Table.
3. All live-cell imaging in Figure 4 should be quantified. In the current figure the authors only provide a single image series and no quantification.

Point-by-point response to the reviewers

Reviewer #1:

In the present study Avilov et al perform a global phosphoproteome analysis of starfish oocytes in metaphase of meiosis I, in control and U0126-treated oocytes. They find that inhibition of the Mos-MAPK pathway affects the meiotic nature of the spindle and Cyclin B translation to drive the second meiotic division. They also found that entry into meiosis I is not affected by U0126, and that fertilization arrest is abolished, however, this has been published previously.

We would like to thank the reviewer for her/his careful and thorough assessment of our work, for the constructive criticism and helpful advice.

Main points:

The experiments are based on previous results published in 2000 in starfish oocyte by Tachibana et al. In this paper, a Mos antisense RNA approach was used, which is much more specific than U0126 to block the Mos-MAPK pathway. Rescue experiments with recombinant GST-Mos were used as controls. Without Mos, 3 cell divisions and DNA replication right after metaphase I were observed most of the time. MAPK activation using phospho-MAPK was analysed and Cyclin A and Cyclin B expression levels were assessed by western blots. The authors need to add these controls (western blots to assess Cyclin B levels, replication immediately after metaphase I, MAPK activity) to make sure that the effect observed is indeed due to Mos-MAPK inhibition alone.

Indeed, prior work by Tachibana and coworkers -- not only the original publication (PNAS, 2000), but also follow up papers by Mori *et al.* (Development, 2006) and Ucar *et al.* (JCS, 2013) -- have characterized Mos-MAPK in starfish oocytes to a substantial detail. These works demonstrated a one-to-one correspondence of phenotypes caused by U0126-treatment, Mos morpholino injection and other perturbations of Mos-MAPK activity. These together established U0126 as an effective and specific tool to inhibit Mos-MAPK activity in starfish oocytes.

However, we agree with the reviewer that our work benefited from additional control experiments. Therefore, we performed Western blots with a phospho-MAPK antibody through stages of meiosis (new Fig. S1C). This confirms that MAPK activity is absent in immature oocytes, but it then increases rapidly after NEBD, and falls back down after fertilization. U0126 treatment completely abolished MAPK activity at all stages of meiosis.

To monitor activation of Cdk1, we used the phospho-Cdk1 (Tyr15) antibody (new Fig. S1C). This confirmed that Cdk1 is activated to a similar extent at the time of NEBD in U0126-treated and untreated oocytes. At this time resolution, Cdk1 appears to remain active through meiosis in control oocytes (we do not resolve the small drop between MI and MII). In U0126-treated oocytes Cdk1 is inactivated earlier, as expected, already at the end of metaphase I.

Furthermore, the additional proteomics experiment we performed during revisions confirmed expected changes in cyclin B levels (new Fig. S3C), while cyclin A could not be detected likely due to the very low expression of the protein in early steps of meiosis in starfish oocytes. Proteins involved in DNA replication were present and slightly upregulated in U0126-treated samples (e.g. the MCM complex, new Fig. S3G), consistent with DNA replication taking place prematurely, as demonstrated earlier (Tachibana *et al.*, PNAS, 2000).

I have some concerns regarding the U0126 concentrations used (20 μ M final). In *Xenopus* oocytes, 50 μ M of U0126 is able to inhibit other kinases than MAPK (monitored by MBP kinase assays) (Dupré et al., 2002, Embo J). Can the authors exclude that 20 μ M of U0126 do not show the same unspecific effect in starfish oocytes? More importantly, and still in *Xenopus*, U0126 decreases Cdk1 activity at GVBD because the Mos-MAPK likely acts as a positive regulator of Cdk1 activation (Gross et al, 2000, Current Biology; Dupré et al., 2022, Embo J). First, this is not observed when oocytes are injected with Mos antisense RNAs to target the Mos-MAPK pathway, and second, it has to be determined whether the same is happening in starfish oocytes. In short, the authors need to determine the concentrations that only affect Mos-MAPK and not Cdk1 at the experimental conditions they are using.

We thank the reviewer for raising this important issue. Firstly, it is important to point out that unlike in *Xenopus* oocytes, in starfish Cdk1 and Mos-MAPK are activated sequentially: Cdk1 is activated upon hormonal stimulation leading to NEBD, while Mos-MAPK activity first appears after NEBD and reaches full activation at around metaphase I (Tachibana, PNAS 2000). In all our experiments we treat oocytes with U0126 at the time of hormone addition. Therefore, if U0126 interfered with activation of Cdk1, we would expect to see a delay in NEBD and subsequent events leading up to metaphase I. As we have shown already on the original Figure 1, at 20 μ M U0126 we do not observe any delay or abnormality in the progression of NEBD, chromosome capture and spindle assembly, indicating normal initial activation of Cdk1.

We now additionally confirmed these observations at different concentrations of U0126, shown on new Fig. S1A, B. While we could confirm that maturation proceeds normally up to 20 μ M concentration, 50 μ M U0126 severely delayed NEBD. Performing Western blots on these samples using the phospho-MAPK antibody and the phospho-Cdk1 (Tyr15) antibody further confirmed that: firstly, up to 20 μ M U0126 concentration Cdk1 activation proceeds normally, whereas 50 μ M U0126 delayed activation of Cdk1. Secondly, these data show that 20 μ M U0126 is necessary to fully inhibit Mos-MAPK activity. This is well consistent with our other experiments in which 20 μ M U0126 was needed to achieve a homogeneous and fully penetrant phenotype.

Providing additional support to these data, Ucar and co-workers (JCS, 2013) performed an elegant rescue experiment, whereby the phenotype caused by 20 μ M U0126 could be rescued by overexpression of constitutively active p90^{RSK}, further evidencing specific inhibition of the Mos-MAPK pathway in starfish oocytes at this concentration. Furthermore, in mouse oocytes use of a similar concentration, 15 μ M U0126 was also found to be necessary for achieving a fully penetrant phenotype (Tong et al., Cell Research, 2003).

Cdk1 and MAPK phosphorylate redundant "SP" sites. The authors performed a phosphoproteome to identify proteins phosphorylated by the Mos-MAPK pathway. These proteins are less phosphorylated in U0126-treated oocytes than in WT oocytes. Since Cdk1 activity may be decreased in U0126, phosphorylation of proteins may be affected by Cdk1 and not MAPK inhibition. For these reasons, Cdk1 activity must be absolutely assayed in metaphase I to make sure that the identified proteins are targeted specifically by the Mos-MAPK pathway.

We agree with the reviewer, but we believe that two separate issues have to be considered here:

Firstly, we understand the reviewer's concern about a direct, off-target inhibition of Cdk1 by U0126. However, as detailed above, we can clearly exclude this effect in starfish oocytes as the initial activation of Cdk1 occurs unperturbed and on schedule even in the presence of 20 μ M U0126.

Secondly, regulation by Mos-MAPK is part of the complex cell cycle signaling network interlinked at multiple points to Cdk1. Therefore, we would in fact expect to see interactions between the two pathways. To gain more insights into this, we added two pieces of new data. First, we performed Western blots with a phospho-Cdk1 (Tyr15) antibody, which revealed that in metaphase I Tyr15 phosphorylation is absent both in DMSO- and U0126-treated samples, indicating that Cdk1 is activated to a similar extent at this stage independently of whether Mos-MAPK is active or inhibited (new Fig. S1C). Second, we extended the analysis using the bioinformatic tool that predicts the kinase responsible for phosphorylation of a given site (<https://kinase-library.phosphosite.org/>). We previously provided a global analysis of all peptides (Fig. 2E). We now extended this analysis by grouping phosphopeptides most likely to be phosphorylated by a given kinase. This analysis revealed that the highest number of peptides are likely targets of the Mos-MAPK-RSK pathway, which is followed by putative Cdk1 targets (Fig. S2B, C). Together, these data suggest that while we primarily affect Mos-MAPK targets, indirectly there is a significant downstream effect on Cdk1 targets as well.

Related to this point, have the author checked the kinase consensus motif of the dephosphorylated phosphosites? Direct Mos-MAPK targets should be SP/TP phosphosites. Since Cdk activity is expected to be affected by U0126 treatment as well, a second check is that the targets are not Cdk sites. Even if the kinase consensus site of Cdk and Mos-MAPK overlap (SP/TP),

Cdk precisely recognises SPx(K/R) and TPx(K/R), which is a first way to discriminate between Cdk and Mos-MAPK targets. The author should provide a logo motif analysis, which is a first quick way to visualize the data.

We included the logo motifs complementing the analysis detailed above (Fig. S2B, C). While these data are generally consistent with the known consensus site of the respective kinase, the logo motifs also visualize the rather subtle difference between MAPK and Cdk1 consensus sequences, distinct from the Mos and RSK consensus.

The statement in the title saying the Mos-MAPK pathway regulates translation is an overinterpretation. There are no data assessing protein translation globally, and in the paper by Tachibana et al there was no effect on Cyclin A and B protein levels when Mos translation was blocked. This discrepancy has to be addressed. Furthermore, in mouse and xenopus oocytes, CPEB phosphorylation promotes the degradation of the protein in metaphase I. What is happening in starfish oocytes is unknown. A western blot may be necessary to make sure that CPEB is still present. They also mentioned that they did a proteomic experiment, in which they do not observe significant changes in protein levels. The data are not shown and a supplementary figure with these data should be included.

Changes in CPEB levels during oocytes meiosis have previously been studied in two different starfish species by Western blotting (Lapasset *et al.*, *Dev. Biol.*, 2005, Ochi *et al.*, *Mol Reprod. Dev.*, 2016). In both species CPEB follows the canonical pattern observed in other species: CPEB is phosphorylated (visualized by a large upshift) after NEBD, followed by degradation later in meiosis. We now performed an additional proteomics time course experiment showing that CPEB peptides can be detected at metaphase I in both control and U0126-treated samples (new Fig. S3E). CPEB levels then decline and at the end of meiosis peptides cannot be detected any more in control oocytes. Interestingly, in U0126-treated oocytes CPEB appears to be partially stabilized (new Fig. S3E).

Cyclin B peptides could also be detected, and as expected, we could observe a small second peak at metaphase II in control oocytes, while this second peak was absent in the U0126-treated sample (new Fig S3C). These data do indicate that Mos-MAPK facilitates Cyclin B synthesis at the MI-MII transition. At the same time, the changes in protein levels are relatively subtle, which would likely remain undetectable by standard, non-quantitative Western blotting. One possible explanation for this subtle change may be local regulation of the protein levels, for example in the proximity of the spindle, which will appear as a small change when whole cell extracts are analyzed. Indeed, we have shown previously that Cyclin B protein forms a gradient along the animal-vegetal axis in starfish oocytes (Bischof *et al.*, *Nat. Comm.*, 2017), and in the future we are planning to explore this spatial regulation further in the context of Mos-MAPK.

Cyclin A was not detected in our samples likely due to the fact that Cyclin A is not present for most part of meiosis in starfish oocytes, and therefore peptides did not pass the statistical threshold in our global analysis.

Finally, we respectfully disagree with the reviewer that our title would be an overinterpretation, as our phosphoproteomic experiment clearly identifies translational regulation as a prominent molecular module affected by Mos-MAPK inhibition. We detect dephosphorylation of a large number of subunits of the CPE-dependent translational regulation complex, and CPEB1 and LARP1 are our top hits (Figs. 2F and 3). However, the reviewer is right in that we are still far from a complete understanding of how Mos-MAPK regulates translation. It will take substantial time and effort to work this out, which we do plan to pursue, but it is not feasible to conclude these experiments within the frames of the revisions. Consistently, we toned down some of our statements throughout the manuscript text.

The article uses various techniques to show that U0126-treated cells enter mitosis after meiosis I. In Figure 1, the decondensation of chromosomes mentioned by the authors is not visible at this resolution. However, the reformation of the lamina is a good argument, but a control is missing showing that in control starfish oocytes there is no lamina in anaphase I, 60 min after NEBD. In Figure 1, the authors also provide a very detailed description of spindle morphology, which is convincing.

We now added insets showing the chromatin channel separately at a larger size on Fig. 1A. We also show a control oocyte at late anaphase I with lamin antibody staining visualizing the absence of lamina at this stage (modified Fig. 1D).

In Figure 4, a different technique is used - looking at the formation of phase-separated condensates of H1-Alexa647 in the context of MO Cyclin B and emetin treatment. Why didn't the authors use the same controls as in Figure 1 for consistency? The lamina should also have been shown. And far more importantly, the author argues that "By contrast, inhibition of Cyclin B translation by morpholino (Fig 4A, B) or by general translation inhibition (Fig. 4D) did not affect the morphology of the MI spindle. This indicates that Mos-MAPK affects MI spindle morphology, and it does so independently of translational regulation". These data are not shown and an analysis similar to Figure 1F is required to draw this conclusion must be provided. Furthermore, this result is very important to demonstrate that the phenotypes are independent of Cyclin B1-Cdk and specific to Mos-MAPK.

We cannot give a strong reason for using different chromatin markers in Fig. 4 and Fig. 1. In previous work we have validated each of these as reliable markers and we use them interchangeably. While H2B gives a 'cleaner' signal highlighting the chromosomes without background, H1 is brighter and the cytoplasmic aggregates serve as a convenient cell cycle marker. While we can label the lamina in live oocytes by overexpressing a laminB-GFP construct, we would be concerned about overexpression artifacts. Overall, we do not think that additional experiments are necessary as we do not expect them to lead us to any different conclusion on this specific point.

However, we completely agree with the reviewer on his/her second point. Therefore, we performed new experiments in which we fixed and stained oocytes at metaphase I (new panels Fig. 4D-F). We treated half of these oocytes with emetine in order to block translation, while the other half of the oocytes we treated with emetine *and* U0126 to block translation *and* the Mos-MAPK pathway. We then quantified this experiment by measuring spindle pole distance and the size of the microtubule asters. This experiment shows that translation inhibition alone does not affect spindle morphology, whereas Mos-MAPK inhibition results in enlarged asters and increased pole-to-pole distance. Thereby, this experiment evidences that Mos-MAPK affects spindle morphology independently of translational regulation.

It seems that there are lots of proteins identified that are involved in DNA replication, when comparing untreated and U0126-treated oocytes. These proteins should be listed in Figure 3B as it may help understanding how Mos-MAPK activity may contribute to inhibition of DNA replication between meiosis I and meiosis II. Can the authors assess protein levels of key proteins involved in replication?

We indeed detected a few differentially phosphorylated peptides of proteins involved in DNA replication. However, we might have been over-excited about these observations, and we should have formulated the corresponding sentence more carefully. We now looked in more detail and we found no statistically significant enrichment of GO terms related to DNA replication and mitotic exit.

However, we find a few specific examples; we detect a differential phosphorylation of the replication licensing factor MCM4, the origin recognition complex subunit ORC3 and the alpha subunit of the DNA polymerase POLA1. These individual examples we now name in the text. In addition, we quantified levels of proteins involved in replication in our proteomics time course experiment (new Fig. S3G). This shows that most components are present through meiosis, however in control oocytes the levels of several components (e.g. MCM complex) drops at the end of meiosis and returns back up during the first mitotic division. Interestingly, U0126 treatment appears to stabilize these proteins (Fig. S3G).

Minor comments:

Regarding the mass spectrometry data and Figure 2, please include the following comments in the MS analysis part in the methods, text and/or figure legends:

- Figure 2A: When were the oocytes treated and for how long before sample collection? the authors also mentioned that they tested cell cycle synchrony (data should be shown in Supp).

U0126 was added simultaneously with the maturation hormone 1-MA, NEBD then commenced in 20-25 minutes and metaphase I was reached at around 60 minutes. We continuously monitored the oocyte population under a dissecting microscope to ensure that the samples are taken at the optimal time point. We then also fixed some of the oocytes from the exact same sample and stained them with Hoechst 33342 to confirm stages based on chromatin morphology. We only documented these results in a tabular form, which we now show in Fig S2A. These details are now included in the Materials and Methods.

- Figure 2B: Numbers in the figure do not match those in the text.

We corrected the numbers. We apologize for the mistake.

- Figure 3C: How many peptides are shown in this hierarchical clustering? Only the 269 high fold change peptides? There are also more than 300 phosphosites shown in the supplementary data. A supplemental dataset with a list of these unique peptides would be appreciated.

P-TMT heatmap was made using the 360 significant, quantified sequence-PTM combinations, summarized by showing the unique peptide sequences. These peptides are now included in Supplemental File 3.

- Figure 3D and 3E: missing description in Materials and Methods

We now provide a description of the kinase prediction tool in the Materials and Methods section.

In general, I was surprised that almost all phosphosites show a decrease in phosphorylation. Only a very small percentage looks phosphorylated. Surely many other substrates including kinases and phosphatases are under control of Mos-MAPK. I would have expected to see a lot of non-direct phosphorylations as well, especially if the U0126 treatment is longer than 5-10 min (this allows enough time for other kinase activation or phosphatase inactivation). Can the author comment on this point?

We do detect some phosphorylation events, but it is true that only 6 peptides pass the significance threshold as compared to 205 dephosphorylated peptides. As explained above, in starfish oocytes Cdk1 and Mos-MAPK are activated sequentially, shortly after hormone addition and after NEBD, respectively. Therefore, while U0126 is present for about an hour before the samples were taken for phosphoproteomics, the target, Mos-MAPK is activated just about 10 minutes before the samples are taken. Thus, these results are consistent with specific inhibition of Mos-MAPK, which is activated shortly before the samples were frozen.

Throughout the figure and the text, please state whether the oocytes are in metaphase I or II and not in metaphase "alone" (the same applies to the other stages). This applies particularly to Figures 5 and 6.

We now consistently label all stages both in the text and figures.

Reviewer #2

Using starfish oocytes as model system, the authors address the function of Mos kinase in oocyte maturation. Mos-MAPK functions are conserved in starfish involving reactivation of CDK1/Cyclin B at the end of MI and a CSF-like arrest, that occurs at the G1 phase. Using U0126, a MEK kinase inhibitor, the authors reveal that loss of the Mos/MAPK pathway cause oocytes to proceed directly from MI to parthenogenetic development. To understand the underlying mechanisms, the authors then perform phospho-proteomics analyses of control and U0126-treated starfish animals. These analyses resulted in the identification of 269 unique phospho-peptides passing stringent filtering criteria. Affected processes were translation, cell cycle and cytoskeleton regulation. Among the translational regulators were CPEB and eIF4ENIF1 and therefore the authors conclude that Mos-MAPK might induce CPE-dependent translation. Specifically, the hypothesize that Mos-MAPK may drive MII by inducing Cyclin B translation. As expected, oocytes not expressing Cyclin B were able to complete MI, but failed to enter MII. Conversely, ectopic CDK1/Cyclin B rescued the failure of U0126-treated oocytes to enter MII. Inhibition of Cyclin B synthesis did not affect the morphology of MI spindles indicating that Mos/MAPK affect MI spindles independently of translational regulation. In fact, the authors find evidence that Mos/MAPK downregulates centrosomal MT nucleation and

thereby contributes to minimizing PB size. In the absence of Mos activity, the MI spindle becomes mitotic-like with astral MTs pushing the spindle away from the cortex.

This manuscript addresses an important aspect of meiosis, i.e., what are the targets of Mos/MAPK in oocytes. As was to be expected from Peter Lenart's group, the experiments were carried out very thoroughly and the data presented are of very high quality. The manuscript will be frequently cited in the future because the role of Mos/MAPK is still unclear. This reviewer thinks that the part on MT dynamics is more convincing than the part on translational regulation, see below for details. This reviewer therefore recommends publication of the paper once the authors have addressed the following points.

We would like to thank the Reviewer for careful assessment of our manuscript and for the helpful comments.

The main concern of this reviewers relates to the translational regulation part:

- Fig. 4: I think it would be very helpful to see the protein levels of cyclin-B in Control and cyclin-B MO-injected oocytes over the course of meiotic maturation (if there is an antibody available).

Unfortunately, we do not any more have a starfish cyclin B antibody to perform this experiment. Both we and Zak Swartz received aliquots of the starfish anti-cyclin B antibody originally developed by the Kishimoto lab. We already used up all of these aliquots. However, Zak Swartz previously could perform this control experiment using a remaining antibody aliquot, and has published these results in his paper showing effective depletion of cyclin B protein after morpholino injection (Swartz, eLife, 2021, Figure 2, supplement 2). We confirmed directly with Zak Swartz that here we use the exact same morpholino and the same depletion protocol, and we observe identical cellular phenotypes resulting from this treatment. In our opinion this provides sufficient evidence that the cyclin B morpholino is an effective means to block synthesis of new cyclin B protein during meiosis. Furthermore, we performed additional experiments using the general translation inhibitor emetine (Fig. 4D-F). These experiments show overall very similar effects to Cyclin B MO injection, further strengthening our conclusions.

- I think it is also not surprising that you need cyclin B synthesis for entry into MII. This has been shown in many species and I think describing the link to the Mos-MAPK pathway is more interesting. However, in my opinion, from the data presented here it is not possible to say if translation of cyclin B is directly affected by the Mos-MAPK pathway. Another possibility could be that cyclin B translation after MI works perfectly fine upon U0126, but Cdk1 is inactivated by inhibitory phosphorylation (Myt1 and Cdc25 are hits in the MS analysis) and the oocytes therefore end up in interphase. This would probably also be rescued by the ectopic Cdk1-CycB used in Fig. 4D (probably depending on how much Cdk1 activity was added compared to endogenous Cdk1 activity - is this known?). In my opinion, this figure would be much more convincing, if the authors could show that translation of a cycB 3'UTR reporter mRNA is affected by inactivation of the Mos-MAPK pathway. The effect could be still indirect, but it would provide more evidence that Mos/MAPK indeed regulates translation. The authors should discuss in more detail the possibility that the effects on translation upon U0126 treatment are indirect or provide data, e.g., in vitro assays, that the effects are direct.

In brief, we agree with the reviewer and we are equally curious about the detailed molecular mechanisms whereby Mos-MAPK regulates translation. However, we think that mapping this complete pathway of translational regulation is beyond the scope of the present manuscript.

In detail to the specific points raised:

- We agree that it is well established that Cyclin B synthesis is needed for MII entry. We state this in the Introduction and refer to relevant previous work in *Xenopus* oocytes;
- We also clearly state (in the Introduction and Discussion) that other mechanisms, such as inactivation of Cdk1 by inhibitory phosphorylation, may act in parallel. Our data does not exclude, it rather indirectly supports that such parallel mechanisms may be at play;
- On the other hand, our phosphoproteomics experiment very clearly shows specific dephosphorylation of many proteins involved in translational regulation upon U0126 treatment. This includes the two top hits showing the largest change, CPEB and LARP1. We believe that this experiment is very carefully controlled, further strengthened by additional controls in the revised version (controls for U0126 on new Fig. S1, as well as the proteomics

time course experiment shown on new Fig. S3). Therefore, we take this result as strong evidence that Mos-MAPK inhibition acts on CPE-dependent translation;

- We refer to previous literature and performed a number of experiments to show that Cyclin B is the main protein that is translated at the MI to MII transition, and that Cyclin B translation is essential for MII entry in starfish oocytes. However, we do not exclude the possibility that Mos-MAPK may regulate translation of other mRNAs as well. We now took care that this is stated clearly: that Mos-MAPK regulates translation is clearly evidenced by our data, but that Cyclin B is the main target of this regulation remains a highly plausible hypothesis based on correlative evidence.
- We invested substantial time and effort to attempt the experiments suggested by the reviewer using 3'UTR reporter constructs. This assay has been successfully used in *Xenopus* and mouse oocytes to study meiotic entry (e.g. Heim *et al.*, Development, 2022, Cheng *et al.*, Nat. Comm., 2024). However, we had to realize that there is a critical difference between these species and MI-to-MII transition in starfish: while the process takes several hours in *Xenopus* and mouse, MI-to-MII transition takes less than 30 minutes in starfish. Over hours, the reporter protein reaches a steady-state providing a reliable measurement (a fact we could confirm by monitoring the baseline level of translation in immature starfish oocytes). Unfortunately, during the rapid progression through meiosis there is insufficient time to reach a steady state in starfish oocytes. On the other hand, in starfish oocytes we are able to continuously monitor the level of the fluorescent reporter, and in principle we should be able to derive the rate of protein synthesis from the change in fluorescence over time. Unfortunately, however, these attempts failed, because the time fluorescent proteins need to mature and become fluorescent is limiting in these assays. Overall, while we are very interested in following up on the detailed mechanisms of translation in starfish oocytes, this will require establishment of new assays (including *in vitro* assays suggested by the reviewer) and will require investment of time and effort beyond what is feasible within the revisions of the present manuscript. Consequently, we substantially re-wrote the relevant sections clearly stating that, while our data provides a strong hypothesis and a specific set of candidate proteins, these will need to be tested in the future in detailed, mechanistic experiments.

- Furthermore, the authors write in M&M that they use the cyclin-B MOs at 500mM final concentration, which is much more than expected. Is this a typo?

Yes, this is a typo, we used 500 μ M just as Swartz et al. We apologize for the mistake.

- Page 5: the authors write „For maintenance of the CSF arrest, Mos-MAPK appears to act on Erp1/Emi2 to prevent Cyclin B degradation by the APC/C (Wu and Kornbluth, 2008). However, the detailed molecular mechanisms remain incompletely understood." There have been follow-up publications working on this mechanism and it has been shown, that p90Rsk phosphorylation of Erp1/Emi2 is required for efficient recruitment of PP2A-B56, which counteracts inhibitory/destabilizing phosphorylations (Isoda et al., 2011 and Hertz et al., 2016).

Thank you! We changed the text accordingly and updated the references.

- Fig. 2B: in the text, the authors write "In total, we quantified 9,035 protein groups, of which 4,716 contained phosphopeptides, 22,822 phospho-peptides in total (Fig. 2B)." To me it is not clear, what the authors refer to as "protein groups"? How do they relate to the 8703 proteins of the left panel in Fig. 2B? It is also not entirely clear to me how the numbers of the left panel in Fig. 2B correlate to the numbers given in the text.

We apologize, there were unfortunately a couple of typos on the figure. We now checked these and made sure all numbers are correct. The overall statistics of the phosphoproteomic experiment is as follows:

Protein groups identified	9035
Phosphoprotein groups identified	3197
Protein groups quantified	7183
Phosphoprotein groups quantified	1252

Identified phosphopeptide sequences	14360
Quantified phosphopeptide sequences	10880
Quantified sequence-PTM combinations	16621
Phosphopeptides quantified showing significant change	211
Sequence-PTM combinations showing significant change	360

We use the term ‘protein group’ as defined by the MaxQuant package to denote proteins, which cannot be unambiguously identified by unique peptides (but have only shared peptides). E.g. if all detected peptides of protein A also belong to protein B, A and B form one protein group. Proteins in each protein group are quantified together. This is especially relevant for our starfish data, as we have rather frequent duplications of proteins in our protein database because the database was derived by combining multiple genomic and transcriptomic resources.

- Fig. 2: the authors describe that many translation regulators are differentially phosphorylated in the U0126-treated oocytes, suggesting that Mos/MAPK are required for translational control during maturation. However, in their MS analysis they found no differentially expressed proteins. Was this expected? In the reference cited (Swartz et al., 2021) there is a small group of proteins that increase in abundance during maturation (already in MI). Are these affected by the U0126 treatment? Later, cyclin B is proposed to be changing in abundance upon U0126 treatment. Are other CPEB-regulated mRNAs expected to be regulated by Mos-MAPK as well? Or is this mechanism in starfish meiosis not used as prominently as in other model organisms? The Mos mRNA was also described to be regulated by CPEB in other species. Do the authors know if the translational upregulation of Mos mRNA itself is regulated by the MAPK pathway in starfish oocytes?

We now performed a proteomics time course experiment to address the points raised by the reviewer:

- We now specify that in the phosphoproteomics experiment we focused on proteins of which phosphopeptides were detected. These proteins showed no significant change in protein levels.
- The proteomics time course experiment showed changes in Cyclin B and CPEB levels similar to findings by Swartz et al, 2021. We now show these on the new Fig. S3C and S3E, respectively. Interestingly, CPEB appears to be partially stabilized in U0126-treated samples. By contrast, Cyclin B levels are slightly reduced in U0126-treated oocytes as compared to DMSO controls, and the small peak at MII is absent. These data together support our model whereby Mos-MAPK acts on CPEB to facilitate Cyclin B synthesis at the MI-MII transition.
- Unfortunately, we did not detect peptides of Imp8 and Pim1, which were other prominent examples in the Schwartz et al, 2021 publication. As in the Swartz paper, we were not able to detect Mos peptides either. We also have no data on the polyA status of the Mos mRNA.

- Page 10: The authors write that "In addition, we detected several changes indicative of an overall signature of M-phase exit, including regulators of DNA replication". Is this expected considering that the U0126-treated samples showed a 92-96% synchronicity in metaphase?

This sentence is an overstatement, which we should have formulated more carefully. We now looked in more detail and we found no overall significant enrichment of GO terms related to DNA replication and mitotic exit. What we find are a few specific examples; for DNA replication we detect a differential phosphorylation of the replication licensing factor MCM4, the origin recognition complex subunit ORC3 and the alpha subunit of the DNA polymerase POLA1. These individual examples we now name in the text.

- Page 11 / Figure 3B: the authors write that it has been shown that eIF4ENIF1 and Maskin/TACC3 are part of the same complexes. To my knowledge this has never been shown (or suggested). Both proteins would also directly compete for eIF4E

Thank you for pointing us to this issue. We have extended and reformulated the relevant sections in the Results as well as in the Discussion including several additional, recent references. We now discuss how unphosphorylated CPEB might be part of a repressive complex containing 4E-T, and potentially eIF4E, DDX6 and LSM14. Homologs of each of these proteins are present in starfish

oocytes, and for some we also detected phosphopeptides. One other protein that we would have expected to be present is Zar1. A Zar1 homolog is present in the starfish genome, but no peptides were detected. We also discuss previous literature demonstrating how in *Xenopus* oocytes CPEB may interact with eIF4E through Maskin/TACC3 interfering with translation initiation.

- Fig. 5B: the authors write that they observe a "slightly increased pole-to-pole distance" in U0126-treated metaphase-I spindles. However, in Figure 1G the half-spindle length in MI was not increased. Although the effect in Fig. 5B is small, can the authors say why there is this discrepancy?

The reason for the discrepancy is that the experiments shown on Figure 1 and Figure 5 were analyzed by different authors and they have quantified spindle morphology using slightly different parameters. On Fig. 1, spindle length refers to the body of the spindle *excluding* the centrosomal area. On Fig. 5, the pole-to-pole distance measurement includes the centrosomal area. For consistency, we now include an additional chart on Fig. 1G showing the pole-to-pole distance.

- Fig. 6C: Similarly, it is not clear to me why in Fig 6C, there is a clear increase in width of the metaphase spindle upon U0126 treatment, whereas there is no difference in MI spindle width in Fig. 1G

We decided to remove the spindle width measurements, as we do not think this is a particularly informative parameter. Again, the discrepancy was caused by the fact that the two authors performing the measurements defined width in slightly different ways (in one case an ellipse was fitted on the body of the spindle, the other was a more direct and manual measurement of the width).

- Fig. 7B: It would help to indicate in the Figure what the conditions on the left and right side are

Thank you, we now added the missing labels.

- Page 18: the authors write "transnational regulation"

We corrected the typo.

- p14, bottom: "...twice in diameter in Mos-MAPK inhibited oocytes as compared to control (Fig. 5F)". It should be 5E.

We corrected the typo.

- p15, first sentence: It should be 5F, not 5G

We corrected the typo.

- p15: "..., which contain much more microtubules and reach a size of 20um towards..." Where are the data shown?

We estimated these values from images like Fig. 6D. We now added labels to Fig. 6C so that readers can easily appreciate the approximate size of the microtubule aster.

- p18: "..., these may use similar molecular mechanisms to maintain arrest by maintaining high levels of Cdk1-Cyclin B activity. This reviewer is puzzled. Does this statement also apply to starfish oocytes, which arrest in G1?

We reformulated this part of the Discussion.

Reviewer #3:

During oogenesis the cell cycle arrests in prophase of meiosis I. Oocyte maturation is promoted by a hormonal signal that triggers the advance through the meiotic divisions. Oocyte activation triggers a series of changes resulting in the increased translation of key cell cycle regulators, such as cyclin B, that promote entry into MI. An important aspect of meiosis is that cells do not fully exit from MI but rapidly progress into MII to perform a reductive division. Extensive work in *Xenopus* and *Drosophila* has documented numerous changes in mRNA polyadenylation and protein translation that are important for progression through the meiotic divisions. In all systems studied CPEB is critical for control of mRNA polyadenylation and progression through meiosis. Avilov et al. describe their work to understand how the Mos kinase contributes to progression through the meiotic divisions. They use live and fixed cell imaging in wild-type and Mos-inhibited oocytes to determine that Mos is required after MI to promote progression into MII. This result is consistent with previous biochemical studies characterizing the timing of Mos activation in starfish oocytes. Additionally, the authors perform phosphoproteomics to identify phosphorylation sites that decrease after Mos inhibition. They identify a large number of altered phosphorylation

sites and speculate about how these phosphorylation events may result in the observed cellular phenotypes. Overall, this study provides a thorough cell biological characterization of the Mos inhibition phenotype in starfish oocytes but does not provide any new insight into the critical substrates or molecular mechanisms. Additionally, the authors need to integrate their results with the extensive literature investigating changes in mRNA polyadenylation and translation during oocyte maturation.

We thank the reviewer for the careful assessment of our work. While we agree with most points, we respectfully disagree in that our work would not provide novel insights. The Mos kinase is a conserved, key regulator of meiosis, and also a potent oncogene if expressed erroneously in other tissues. Mapping the conserved molecular pathways downstream of Mos-MAPK is therefore broadly relevant both for the fields of reproduction biology and cancer research. To our knowledge, we report here the first phosphoproteomic analysis to systematically identify downstream targets of the Mos-MAPK pathway in oocytes. We consider this analysis highly successful, as it identified a low number of functional modules with many proteins/subunits in each module phosphorylated in a Mos-MAPK-dependent manner. Importantly, we were able to directly relate the function of these modules to the observed cellular phenotypes. Specifically, we identified a set of proteins involved in cytoskeletal regulation involved in the asymmetric meiotic divisions. Secondly, our analysis identified CPEB-dependent translation as the target of Mos-MAPK.

We believe that the reviewer slightly misunderstood our primary aim: we aimed to understand the cellular phenotypes caused by Mos-MAPK inhibition and to correlate these with changes to the phosphoproteome. We were delighted to see consistent changes on many proteins involved in translational regulation, together supporting the hypothesis that Mos-MAPK may drive meiosis II by timely translation of Cyclin B.

It was not our goal to dissect the mechanism of translational regulation in starfish oocytes in detail. Indeed, this process has been already extensively studied in *Xenopus* oocytes, as well as in other species. The high degree of conservation of the proteins involved therefore allows us to put forward specific hypotheses based on this already existing knowledge. Following the reviewer's suggestion, we extended these sections of the manuscript and included many additional references.

We also would like to explain our apparent ignorance of previous work on *Xenopus* and *Drosophila* oocytes. We are very much aware of the extensive literature on translational regulation, including the greatly impressive seminal work by Joel Richter's and Raúl Méndez's laboratories. However, in the initial submission we made the choice to limit ourselves to works that specifically concern Mos-MAPK and its effects on the regulation of MI-to-MII transition. Therefore, we cited fewer works on *Drosophila*, as in this species Mos-MAPK has a minor role in meiotic regulation (that may be generally the case for protostome species). In *Xenopus*, the majority of works focus on the first wave of translation directly induced by the maturation hormone. In starfish, this initial response to the hormone does not involve translation of new protein. In the revised version of the manuscript we now include several additional references and cover the topics, which were left out in our original submission.

Major Points

1. In Figures 2 and 3 the authors report a proteomic analysis of phosphorylation sites that increase and decrease after Mos inhibition. This data is a useful starting point to understand the molecular functions of Mos during meiosis. The major concern to this area of the manuscript is that the authors do not pursue these results further. The authors include speculation about the potential functions of these phosphorylation sites, but none are tested individually and none are validated using an orthogonal method. In order for this work to be an important contribution to the field it would be necessary to understand how some of the individual phosphorylation sites lead to the observed cellular phenotypes. Since there is a huge body of work examining mRNA polyadenylation during oocyte maturation (see below) that could be an interesting starting point.

We respectfully disagree with the reviewer that our work represents no important contribution to the field. Firstly, we perform for the first time a phosphoproteomics analysis of oocytes with the Mos-MAPK pathway inhibited. This analysis provided not just an accidental set of differentially phosphorylated proteins, but defined specific molecular modules fully consistent with the observed cellular phenotypes. This is in our opinion a major advance toward understanding core conserved functions of Mos-MAPK in regulation of oocyte meiosis.

We agree with the reviewer that the logical next step will be to follow up on specific proteins and phosphosites identified by phosphoproteomics; validate them in perturbation experiments *in vivo* and/or dissect molecular mechanisms in detail in *in vitro* assays. However, we feel that this is not the aim and is beyond the scope of the present study. Additionally, starfish oocytes may not be the ideal model system for such molecular studies -- *Xenopus* oocytes and egg extracts, *in vitro* systems and cell culture models may be better alternatives.

2. There is a tremendous amount of literature about the molecular changes that mediate oocyte maturation in *Xenopus* and *Drosophila* that the authors have not cited or discussed in their work. For example, there are intricate feedback loops that govern CPEB-mediated mRNA polyadenylation and protein translation in *Xenopus* (PMID: 20531391, 18536713, 18385675, 18267074, 31896558) and extensive characterization of the same processes in *Drosophila* (27474798, 25349405, 24882012). Since a major claim of this manuscript is that the translation regulation module is the key to the MI to MII transition it is essential that the authors cite and discuss these studies.

We thank the reviewer for these suggestions. We now extensively re-wrote the Discussion section and included all these references related to the *Xenopus* work. We also cite some of the *Drosophila* work, as they show that CPEB-mediated translational regulation is widely conserved across animal species. On the other hand, in terms of oogenesis and oocyte meiosis *Drosophila* and generally protostome species diverged rather far from the deuterostome lineage (that include the echinoderm starfish and vertebrates). Therefore, we think the details are less relevant to our work. Instead, we focused on studies in oocytes of *Xenopus* and mouse.

3. The authors speculate that Mos directly phosphorylates CPEB to mediate cyclin B translation between MI and MII. However, this is in contrast to the observations in *Xenopus* and mouse oocytes where Aurora-A (Eg2) is responsible for CPEB phosphorylation (11526086, 11106762, 10749216). The authors need to cite these studies and discuss these previous results in the context of their observation that Aurora-A kinases are repressed after Mos inhibition.

We now looked into our data in more detail and discuss the references above with regard to the role of Aurora A. Firstly, the starfish genome does not contain duplications typical to vertebrates, and therefore it contains a single CPEB protein and a single Aurora kinase (that alone carries out functions of vertebrate Aurora A and B kinases, Abe *et al.*, J. Cell Science, 2010). The region of CPEB at which Aurora phosphorylates vertebrate CPEB1 appears to be conserved in starfish (see figure below), however we detect no phosphorylation of any of the neighboring sites by phosphoproteomics. The main site that we see differentially phosphorylated upon U0126 treatment is S32 that is not a conserved site, however it lies near the T22 site that has been found in *Xenopus* oocytes to be phosphorylated by Mos-MAPK (Keady *et al.*, J. Cell Science, 2007).

Consensus		* . . . *		* * * *			
		---	X D X R X I	---	I D S R S S S P S D S D T S G F S S G S	---	
CPEB1 human	170	---	L D T R P I	---	L D S R S S S P S D S D T S G F S S G S	---	195
CPEB1 mouse	169	---	L D T R P I	---	L D S R S S S P S D S D T S G F S S G S	---	194
CPE1B Xenopus	172	---	L D S R S I	---	L D S R S S S P S D S D T S G F S S G S	---	197
CPEB Patiria	319	P G S S	D S S I	W E H V F S P I E R P G A R D	T R S S S P T D S D T S G I S S A S	A S S G	364

Generally, this is consistent with the model in *Xenopus*, whereby Aurora phosphorylation of CPEB occurs early, shortly upon progesterone addition, whereas the later hyperphosphorylation is mediated by Cdk1 and Mos-MAPK. I.e. in our samples we see no evidence for Aurora phosphorylation, and inhibition of Aurora in starfish oocytes has no effect that would suggest a role in translational regulation (Abe *et al.*, J. Cell Science, 2010). Instead, our interpretation is that what we see is Mos-MAPK's contribution to CPEB hyperphosphorylation, which has been shown to release CPEB from the repressive RNP complexes (Thom *et al.*, Biochem J, 2003)

We now included these details in the manuscript in the Results and Discussion citing those references mentioned by the reviewer as well as others.

4. In Figure 2E the authors describe families of kinases that exhibit lower activity after Mos inactivation. It would be useful to provide information about how many phosphosites are direct targets of Mos vs. indirectly regulated targets.

To address this point, we extended the analysis using a prediction tool to identify putative human protein kinases responsible for phosphorylation following the reviewer's suggestion. We now report the number of peptides that are phosphorylated by Cdk1, Mos, MAPK and RSK kinases with the highest probability on the new Fig. S2B and C.

Minor Points

1. The plot in Figure 2F is completely unreadable. The authors should simplify this plot by removing the gene names and only including the names of a few key proteins that they would like to highlight.

We see the point of the reviewer, but we think that most readers rather read the digital version of the publication, in which case the protein names can be easily read by zooming in into the figure.

2. The phosphopeptides present in Figure 3A are hardly discussed in the text and do not meaningfully contribute to the presentation of the data. This data would be more appropriate for a Supplemental Table.

As suggested, we moved this table to the new Fig. S2D.

3. All live-cell imaging in Figure 4 should be quantified. In the current figure the authors only provide a single image series and no quantification.

We now added the sample number (n) to all experiments shown on Figure 4. In addition, we performed a new experiment to show that block of translation by emetine alone has no effect on spindle morphology, implying that Mos-MAPK acts on the cytoskeleton and translation independently (new panels Fig. 4D-F). We imaged many oocytes at high resolution for this experiment and we provide a detailed quantification and appropriate statistical analysis of spindle morphology.

March 17, 2025

Re: JCB manuscript #202312140R

Peter Lenart
Max Planck Institute for Multidisciplinary Sciences

Dear Dr. Lenart,

Thank you for submitting your revised manuscript entitled "Phosphoproteomics identifies targets of Mos-MAPK regulating translation and the spindle in oocytes." The manuscript has been seen by the original reviewers whose full comments are appended below.

You will see that Reviewer #2 states that their concerns have been addressed. However, the other two Reviewers, while noting that the manuscript is improved, still feel that the study does not provide a direct connection between Mos-MAPK phosphorylation and regulation of translation during oocyte meiosis. This was a major concern in the first review round and given the claims made by the title and the abstract, we cannot proceed further with this study without more definitive experimental evidence supporting this proposed link.

Our general policy is that papers are considered through only one revision cycle; however, given interest in the topic, we are open to one additional round of revision, if you are able to provide additional data that demonstrates a more direct link between Mos-MAPK phosphorylation and regulation of translation. We of course understand that this may not be feasible, in which case it may be best to pursue publication at another journal in order to expedite publication of this data.

Regardless of how you choose to proceed, we hope that the comments below will prove constructive as your work progresses. We would be happy to discuss the reviewer comments further once you've had a chance to consider the points raised in this letter. You can contact the journal office with any questions at cellbio@rockefeller.edu.

Thank you again for thinking of JCB as an appropriate place to publish your work.

Sincerely,

William Bement, PhD
Monitoring Editor
Journal of Cell Biology

Dan Simon, PhD
Scientific Editor
Journal of Cell Biology

Reviewer #1 (Comments to the Authors (Required)):

The authors have replied to all our comments, and the manuscript has greatly improved. However, the conclusion that Mos/MAPK is necessary for translational control of Cyclin B at entry into meiosis II is not fully supported by the data provided. In my opinion, the manuscript should be published, but without putting so much emphasis on translational control, if there are too many technical obstacles to study this in this model system.

The fact that there is no drop in Cyclin B protein levels in UO126 treated oocytes suggests that there is no degradation of Cyclin B, and not that there is a problem in translation of Cyclin B after metaphase I (Figure S3C). Moreover, since there is no decrease and slight accumulation of Cyclin B in controls, no conclusion can be drawn concerning the turnover of Cyclin B in meiosis II. In Figure 1C, inhibitory phosphorylation of Cdk1 is detected, maybe explaining why these oocytes can exit meiosis I and then enter G1. The rescue experiment by injecting active Cyclin B-Cdk1 complexes does not demonstrate that the Mos/MAPK pathway is required for translation of Cyclin B, but instead, that there is no active Cyclin B-Cdk1. (Maybe the kinase phosphorylating Cdk1 remains active or is activated while oocytes exit meiosis I?) To show that the Mos/MAPK pathway is required for translation, the authors would need reporter constructs to show presence and absence of translation, in unchallenged and UO126 treated oocytes. If this is technically not possible, the authors have to find other ways to address this issue, or refrain from concluding that the Mos/MAPK pathway regulates Cyclin B translation at entry into meiosis II. Another point concerns CPEB regulation by Mos/MAPK. While the authors propose that CPEB is stabilized in UO126-treated

oocytes, the protein intensity observed in metaphase I is lower than in controls. However, as the oocytes progress directly in G1, the level of CPEB appears to decrease to ultimately reach similar level in both UO126 or controls. Given CPEB's function as an activator of translation, its reduced expression in UO126 could account for the initially lower level of cyclin B in this condition, thereby affecting overall Cyclin B translation already during meiosis I.

Reviewer #2 (Comments to the Authors (Required)):

The authors have done a great job in addressing all my concerns. I therefore recommend publication of the manuscript.

Reviewer #3 (Comments to the Authors (Required)):

This is a revision of a previous submission in which the authors use imaging and proteomics to investigate the function of the Mos kinase in starfish oocyte meiosis. The authors report that inhibition of Mos leads to a normal M1, failure to enter into M2, and entry into parthenogenic mitosis. Many of the cytological characterizations of the Mos inhibition phenotype have been previously reported in starfish and other systems. The novelty of the manuscript is that the authors use phospho-proteomics to identify a long list of phospho-peptides that change in M1 after Mos inhibition. These altered phospho-peptides are predicted to be targets of the Mos pathway, cdk1, and Aurora kinase. The differentially phosphorylated peptides suggest that Mos may control mRNA polyadenylation, protein translation and some aspects of spindle assembly. Overall, the data in this paper is strong and of very high quality. However, the authors have not made any direct connection between altered protein phosphorylation and the observed phenotypes. The authors speculate on how phosphorylation could affect specific processes, but no direct tests of their models are presented. The paper presents an important list of likely Mos targets during starfish oocyte meiosis but does not provide data supporting the strong conclusions in the manuscript. In the revision the authors addressed my concerns about putting their results into the proper literature context but did not address any of my concerns about making a direct connection between phospho-sites and the observed phenotypes. I believe that this work requires significant additional experiments to be appropriate for publication in the JCB.

Major concerns

1. The authors have made no effort to test the function of any identified phospho-sites and state that this work is beyond the scope of this manuscript. This absence of experimental validation is especially evident when discussing the role of Mos in cytoplasmic polyadenylation and translation. The authors speculate that Mos is required to phosphorylate CPEB to promote cyclin B translation required to drive entry into M2. This speculation is highlighted by this sentence in the abstract:

"This revealed CPE-mediated mRNA polyadenylation as a target of Mos-MAPK, and our data suggest that this mechanism regulates cyclin B translation to drive the second meiotic division."

However, several pieces of data in the manuscript do not fit with this model and no aspect of this model is directly tested. First, Mos oocytes fail to progress into M2 but do enter into the first mitosis. Cyclin B MO or translation inhibition recapitulate a failure to enter into M2, but do not enter into the first mitosis. Why is Mos kinase required only for Cyclin B translation for M2 entry but not mitosis? Second, in their time resolved proteomics (S3D) cyclin B levels do not decrease in interkinesis. The authors speculate as to these causes, but this is not consistent with their model. Third, CPEB persists in UO126-treated oocytes. Is this related to phosphorylation? Finally, although the authors state that polyadenylation and translation are regulated by Mos but they present no evidence that either of these processes are altered in UO126-treated oocytes.

In order for this to provide important biological insight into the role of Mos in oocyte meiosis the authors need to provide evidence to support their speculation of the role of Mos phosphorylation. Without this validation this manuscript is a valuable list of phosphorylation sites and speculation as to function.

2. The authors present data from a proteomics time course in the supplemental data, but never properly introduce this experiment in the main text of the results. This is likely a valuable experiment and deserves to be discussed in the main results section.

List of changes to the manuscript #202312140R "Phosphoproteomics identifies targets of Mos-MAPK in cell cycle and asymmetric divisions of oocytes"

Formatting changes

As requested, we restructured our manuscript in order to (i) tone down claims related to translational regulation, and (ii) shortened it to match JCB's Report format. Therefore, we made the following changes:

Figure 1

- We moved the P-MAPK and P-Cdk1 Western blots to the main Fig. 1A (previously Fig. S1)
- We moved the immunofluorescence spindle data from Fig. 1E-G to supplemental Fig. S1C-E
- We moved the Cdk1-Cyclin B injection experiment to Fig. 1F (previously Fig. 4C)

Figure 2

- We changed slightly the schematics of the experimental outline (Fig. 2A)
- We moved the overall statistics presented on Fig. 2B to the text
- We added a schematic of the proteomics time course experiment and moved the heatmap and profiles of key proteins to Fig. 2E-G (previously shown in supplemental Fig. S2A-E)

Figure 3

- We moved the volcano plot from previous Fig. 2F to Fig. 3A

Figure 4

- The new Fig. 4 shows all panels combined from previous Figs. 5 and 6.

Figure 5

- We moved panels of the previous Fig. 4A and D-F to the new Fig. 5 showing translation inhibition experiments by Emetine
- Of the other panels of previous Fig. 4, the Cdk1-Cyclin B injection experiment (Fig. 4C) we moved to Fig. 1F, and the Cyclin B morpholino experiment (Fig. 4B), we moved to the supplement (Fig. S4)
- Fig. 5E shows a schematic of Mos-MAPK mediated changes to meiotic spindle organization. We removed the schematics presenting our hypothetical model of Mos-MAPK's role in translational regulation.

With these modifications we focus our conclusions on showing that: (i) Mos-MAPK inhibition prevents entry to meiosis II that can be rescued by artificial reactivation of Cdk1-Cyclin B; (ii) Mos-MAPK's effect on translational regulation and spindle organization are independent of one another. We completely removed our claims regarding the hypothesis that Mos-MAPK regulates translation of Cyclin B (which we nevertheless find an attractive hypothesis that we plan to pursue in future work).

Response to the reviewers

Reviewer #1 (Comments to the Authors (Required)):

The authors have replied to all our comments, and the manuscript has greatly improved. However, the conclusion that Mos/MAPK is necessary for translational control of Cyclin B at entry into meiosis II is not fully supported by the data provided. In my opinion, the manuscript should be published, but without putting so much emphasis on translational control, if there are too many technical obstacles to study this in this model system.

The fact that there is no drop in Cyclin B protein levels in UO126 treated oocytes suggests that there is no degradation of Cyclin B, and not that there is a problem in translation of Cyclin B after metaphase I (Figure S3C). Moreover, since there is no decrease and slight accumulation of Cyclin B in controls, no conclusion can be drawn concerning the turnover of Cyclin B in meiosis II. In Figure 1C, inhibitory phosphorylation of Cdk1 is detected, maybe explaining why these oocytes can exit meiosis I and then enter G1. The rescue experiment by injecting active Cyclin B-Cdk1 complexes does not demonstrate that the Mos/MAPK pathway is required for translation of Cyclin B, but instead, that there is no active Cyclin B-Cdk1. (Maybe the kinase phosphorylating Cdk1 remains active or is activated while oocytes exit meiosis I?) To show that the Mos/MAPK pathway is required for translation, the authors would need reporter constructs to show presence and absence of translation, in unchallenged and UO126 treated oocytes. If this is technically not possible, the authors have to find other ways to address this issue, or refrain from concluding that the Mos/MAPK pathway regulates Cyclin B translation at entry into meiosis II.

We substantially toned down our claims regarding translational control of Cyclin B, as suggested by the reviewer. Therefore, we removed the word 'translation' from the title, we changed the abstract, substantially reorganized the relevant Results section, and almost completely removed the related discussion. We are nevertheless very interested in exploring this hypothesis further, and therefore plan to establish the required assays in the starfish oocyte model system -- this, however, is expected to take substantial time and effort, much beyond what would be feasible in frames of a revision.

Another point concerns CPEB regulation by Mos/MAPK. While the authors propose that CPEB is stabilized in UO126-treated oocytes, the protein intensity observed in metaphase I is lower than in controls. However, as the oocytes progress directly in G1, the level of CPEB appears to decrease to ultimately reach similar level in both UO126 or controls. Given CPEB's function as an activator of translation, its reduced expression in UO126 could account for the initially lower level of cyclin B in this condition, thereby affecting overall Cyclin B translation already during meiosis I.

We thank the reviewer for pointing this out. We changed the text accordingly to mention this exciting possibility (Lines 198-200 on Page 7).

Reviewer #2 (Comments to the Authors (Required)):

The authors have done a great job in addressing all my concerns. I therefore recommend publication of the manuscript.

We thank the reviewer for his/her positive evaluation of our work!

Reviewer #3 (Comments to the Authors (Required)):

This is a revision of a previous submission in which the authors use imaging and proteomics to investigate the function of the Mos kinase in starfish oocyte meiosis. The authors report that inhibition of Mos leads to a normal M1, failure to enter into M2, and entry into parthenogenic mitosis. Many of the cytological characterizations of the Mos inhibition phenotype have been previously reported in starfish and other systems. The novelty of the manuscript is that the authors use phospho-proteomics to identify a long list of phospho-peptides that change in M1 after Mos inhibition. These altered phospho-peptides are predicted to be targets of the Mos pathway, cdk1, and Aurora kinase. The differentially phosphorylated peptides suggest that Mos may control mRNA polyadenylation, protein translation and some aspects of spindle assembly. Overall, the data in this paper is strong and of very high quality. However, the authors have not made any direct connection between altered protein phosphorylation and the observed phenotypes. The authors speculate on how phosphorylation could affect specific processes, but no direct tests of their models are presented. The paper presents an important list of likely Mos targets during starfish oocyte meiosis but does not provide data supporting the strong conclusions in the manuscript. In the revision the authors addressed my concerns about putting their results into the proper literature context but did not address any of my concerns about making a direct connection between phospho-

sites and the observed phenotypes. I believe that this work requires significant additional experiments to be appropriate for publication in the JCB.

Major concerns

1. The authors have made no effort to test the function of any identified phospho-sites and state that this work is beyond the scope of this manuscript. This absence of experimental validation is especially evident when discussing the role of Mos in cytoplasmic polyadenylation and translation. The authors speculate that Mos is required to phosphorylate CPEB to promote cyclin B translation required to drive entry into M2. This speculation is highlighted by this sentence in the abstract:

"This revealed CPE-mediated mRNA polyadenylation as a target of Mos-MAPK, and our data suggest that this mechanism regulates cyclin B translation to drive the second meiotic division."

However, several pieces of data in the manuscript do not fit with this model and no aspect of this model is directly tested. First, Mos oocytes fail to progress into M2 but do enter into the first mitosis. Cyclin B MO or translation inhibition recapitulate a failure to enter into M2, but do not enter into the first mitosis. Why is Mos kinase required only for Cyclin B translation for M2 entry but not mitosis? Second, in their time resolved proteomics (S3D) cyclin B levels do not decrease in interkinesis. The authors speculate as to these causes, but this is not consistent with their model. Third, CPEB persists in U0126-treated oocytes. Is this related to phosphorylation? Finally, although the authors state that polyadenylation and translation are regulated by Mos but they present no evidence that either of these processes are altered in U0126-treated oocytes.

In order for this to provide important biological insight into the role of Mos in oocyte meiosis the authors need to provide evidence to support their speculation of the role of Mos phosphorylation. Without this validation this manuscript is a valuable list of phosphorylation sites and speculation as to function.

To address criticism, we substantially toned down our claims regarding translational control of Cyclin B. Therefore, we removed the word 'translation' from the title, we changed the abstract, substantially reorganized the relevant Results section, and almost completely removed the related discussion. In the reformatted and shortened version of the manuscript, we generally shifted the focus towards Mos-MAPK's roles in regulation of microtubule spindle organization.

At the same time, we fully share the reviewer's excitement about mechanisms of translational regulation in oocytes, and we are very interested in exploring our hypothesis further. Therefore, we plan to establish the required assays in the starfish oocyte model system -- this, however, is expected to take substantial time and effort, much beyond what would be feasible in frames of a revision.

2. The authors present data from a proteomics time course in the supplemental data, but never properly introduce this experiment in the main text of the results. This is likely a valuable experiment and deserves to be discussed in the main results section.

We now included several panels in the main Fig 2. documenting the proteomics time course experiment, and we added details to the main text (Lines 160-168 on Page 6).

August 7, 2025

RE: JCB Manuscript #202312140RR

Peter Lenart
Max Planck Institute for Multidisciplinary Sciences

Dear Peter,

Thank you for submitting your revised manuscript entitled "Phosphoproteomics identifies targets of Mos-MAPK in cell cycle and asymmetric divisions of oocytes". We would be happy to publish your paper in JCB pending final revisions necessary to meet our formatting guidelines (see details below). Many thanks for your patience and for submitting this interesting study to JCB.

A. MANUSCRIPT ORGANIZATION AND FORMATTING:

- 1) Figure formatting: Scale bars must be present on all microscopy images, including inset magnifications. Molecular weight or nucleic acid size markers must be included on all gel electrophoresis. Please add a scale bar for magnifications in Fig. 1B and molecular weight markers for blots in 1A & S1B.
- 2) Statistical analysis: Statistical methods should be explained in full in the materials and methods. For figures presenting pooled data the statistical measure should be defined in the figure legends. Please also be sure to indicate the statistical tests used in each of your experiments (both in the figure legend itself and in a separate methods section) as well as the parameters of the test (for example, if you ran a t-test, please indicate if it was one- or two-sided, etc.). Also, if you used parametric tests, please indicate if the data distribution was tested for normality (and if so, how). If not, you must state something to the effect that "Data distribution was assumed to be normal but this was not formally tested."
- 3) Title: To convey the advance in a more active voice we suggest revising the title to the following:
"Phosphoproteomic identification of Mos-MAPK targets in oocyte cell cycle and asymmetric divisions".
- 4) Please add the following information to the Materials and methods section:
 - a. The type of membrane used for immunoblotting.
 - b. Catalog numbers or vendor IDs for:
 - anti-mouse secondary nanobody with Atto-488 (NanoTag)
 - donkey Fab anti-mouse IgG AlexaFluor594 (Jackson)
 - anti-p44/42 MAPK (Erk1/2) antibody (Cell Signaling Technology)
 - anti-phospho-cdc2(Tyr15) antibody (Cell Signaling Technology)
 - anti-mouse CF770, anti-mouse CF680, anti-rabbit CF680, & anti-rabbit CF770 secondary antibodies
- 5) Tables, like figures, should be provided as individual, editable files. It may be best to convert figure S3 into a table.
- 6) Conflict of interest statement: JCB requires inclusion of a statement in the acknowledgements regarding competing financial interests. If no competing financial interests exist, please include the following statement: "The authors declare no competing financial interests." If competing interests are declared, please follow your statement of these competing interests with the following statement: "The authors declare no further competing financial interests."
- 7) ORCID IDs: ORCID IDs are unique identifiers allowing researchers to create a record of their various scholarly contributions in a single place. Please note that ORCID IDs are required for all authors. At resubmission of your final files, please be sure to provide your ORCID ID and those of all co-authors.
- 8) Source Data Figures should be provided as individual PDF files (one file per figure). Authors should endeavor to retain a minimum resolution of 300 dpi or pixels per inch. Please review our instructions for export from Photoshop, Illustrator, and PowerPoint here: <https://rupress.org/jcb/pages/submission-guidelines#revised>

B. FINAL FILES:

Please upload the following materials to our online submission system. These items are required prior to acceptance. If you

have any questions, contact JCB's Managing Editor, Lindsey Hollander (lhollander@rockefeller.edu).

****It is JCB policy that if requested, original data images must be made available to the editors. Failure to provide original images upon request will result in unavoidable delays in publication. Please ensure that you have access to all original data images prior to final submission.****

****The license to publish form must be signed before your manuscript can be sent to production. A link to the electronic license to publish form will be sent to the corresponding author only. Please take a moment to check your funder requirements before choosing the appropriate license.****

Thank you for your attention to these final processing requirements. Please revise and format the manuscript and upload materials within 7 days. If you need an extension for whatever reason, please let us know and we can work with you to determine a suitable revision period.

Thank you again for this interesting contribution, we look forward to publishing your paper in Journal of Cell Biology.

Sincerely,

William Bement, PhD
Monitoring Editor
Journal of Cell Biology

Dan Simon, PhD
Scientific Editor
Journal of Cell Biology